# Coupling radiative, conductive and convective heat-transfers in a single Monte Carlo algorithm: A general theoretical framework for linear situations

**Jean Marc Tregan[1], Jean Luc Amestoy[1], Megane Bati[16], Jean-Jacques Bezian[8], Stéphane Blanco[1]\*, Laurent Brunel[4], Cyril Caliot[5], Julien Charon[9], Jean-Francois Cornet[3], Christophe Coustet[14], Louis d'Alençon[1,2], Jeremi Dauchet[3], Sebastien Dutour[1], Simon Eibner[8], Mouna El Hafi[8], Vincent Eymet[14], Olivier Farges[10], Vincent Forest[14], Richard Fournier[1], Mathieu Galtier[11], Victor Gattepaille[3], Jacques Gautrais[1,7], Zili He[8], Frédéric Hourdin[2], Loris Ibarrart[13], Jean-Louis Joly[1], Paule Lapeyre[12], Pascal Lavieille[1], Marie-Helene Lecureux[6], Jacques Lluc[1], Marc Miscevic[1], Nada Mourtaday[6], Yaniss Nyffenegger-Péré[1], Lionel Pelissier[6], Lea Penazzi[10], Benjamin Piaud[14], Clément Rodrigues-Viguier[1], Gisele Roques[1], Maxime Roger[11], Thomas Saez[15], Guillaume Terrée[11], Najda Villefranque[17], Thomas Vourc'h[3], Daniel Yaacoub[3]**

**1** LAPLACE, Université de Toulouse, CNRS, INPT, UPS, Toulouse, France, **2** LMD/IPSL, Sorbonne Université, CNRS, École Polytechnique, ENS, Paris, France, **3** Université Clermont Auvergne, Clermont Auvergne INP, CNRS, Institut Pascal, Clermont-Ferrand, France, **4** PhotonLyx Technology S.L., Santander, Spain, **5** CNRS, UPPA, E2S, LMAP, 1 Allée du Parc Montaury, Anglet, France, **6** Inspe TOP, UMR EFTS, Université de Toulouse, Toulouse, France, **7** CRCA, CBI, Université de Toulouse, CNRS, Toulouse, France, **8** Université de Toulouse, Mines Albi, UMR 5302 - Centre RAPSODEE, Campus Jarlard, Albi, CT, France, **9** ESTACA West Campus, Rue Georges Charpak, Laval, France, **10** Université de Lorraine, CNRS, LEMTA, Vandœuvre-lès-Nancy, France, **11** Univ. Lyon, CNRS, INSA-Lyon, Université Claude Bernard Lyon 1, CETHIL UMR5008, Villeurbanne, France, **12** Department of Mechanical and Mechatronics Engineering, University of Waterloo, Waterloo ON, Canada, **13** Cnes, Toulouse, France, **14** Méso-Star, Longages, France, **15** LTZ Electronique, Saint Laurent du Var, France, **16** IRIT, Université de Toulouse, CNRS, UPS, Toulouse, France, **17** Centre National de Recherches Météorologiques, UMR 3589 CNRS, Météo France, Toulouse, France

\* stephane.blanco@laplace.univ-tlse.fr

**Data Availability Statement:** All relevant data are within the paper and its Supporting information files.

## Abstract

It was recently shown that radiation, conduction and convection can be combined within a single Monte Carlo algorithm and that such an algorithm immediately benefits from state-of-the-art computer-graphics advances when dealing with complex geometries. The theoretical foundations that make this coupling possible are fully exposed for the first time, supporting the intuitive pictures of continuous thermal paths that run through the different physics at work. First, the theoretical frameworks of propagators and Green's functions are used to demonstrate that a coupled model involving different physical phenomena can be probabilized. Second, they are extended and made operational using the Feynman-Kac theory and stochastic processes. Finally, the theoretical framework is supported by a new proposal for an approximation of coupled Brownian trajectories compatible with the algorithmic design required by ray-tracing acceleration techniques in highly refined geometry.

**Funding:** This work received financial support from the Agence Nationale de la Recherche (French National Agency for Research), https://anr.fr/, (ANR project HIGH-TUNE ANR-16-CE01-0010, ANR project MC2 ANR-21-CE46-0013 and ANR project MCG-RAD ANR-18-CE46-0012), from Region Occitanie (Projet CLE EDSTAR), from French government research program "Investissements d'Avenir" through the IDEX-ISITE initiative 16-IDEX-0001 (CAP 20-25), the IMobS3 Laboratory of Excellence (ANR-10-LABX-16-01) and the SOLSTICE laboratory of Excellence (ANR-10-LABX-22-01). The funders had no role in study design, data collection and analysis, decision to publish, or preparation of the manuscript.

**Competing interests:** The authors have declared that no competing interests exist.

# 1 Introduction

## 1.1 The proposition

In corpuscular physics, radiative transfer can be described in the framework of the linear Boltzmann theory for photon transport and this leads quite naturally to path-space formulations. The Monte Carlo (MC) method then allows to estimate the quantity of interest by sampling the paths according to their probability law so as to identify the contributing boundary conditions or sources. This statistical method is the only one able to provide an unbiased estimate along with its statistical uncertainty. Furthermore, as MC methods are able to handle complex integration domain (spatial, angular, spectral. . .), it is widely used as a reference method. Finally, the insensitivity of the method to the size and level of refinement of the geometric scenes and volumic heterogeneities (see, for example, [1–4]) has further widened its applicability, up to the industrial scale.

In the present paper, we give the theoretical basis for extending MC methods to problems including coupled linear heat transfers. We aim at providing a complete framework for describing, in a single MC algorithm, the coupled energy transfers as conductive, convective and radiative paths, while retaining the flexibility of standard MC methods applied to linear transport theory.

Our standpoint being here theoretical, we will not consider any implementation associated with a specific system in practice. Starting from earlier developments [5], successful implementations have already been reported for several practical heat transfer applications. For instance, using this framework to model detailed thermal transfers in cities has been proposed as a way to improve climate services [6]. Applied works are also in progress in complex cooling systems design such as power electronic systems that are cooled by air or two-phase exchangers [7, 8], thermal receivers in concentrated solar power plants [9, 10], electric motors, or thermal housing issues. . . On a more theoretical level, [11–13] explores the benefits of propagating the adjoint model and proposes suitable computational implementations: it is a matter of using properties of MC methods that allow to estimate source influences in the sense of Green's theory. Considering implementation and performance of statistical methods, [14, 15] demonstrate computational insensitivities to geometrical complexity on engineering applications in porous media. These works are mostly based on a free-licensed library (the Stardis project [16]) which implements most of the elements presented below. Although their common theoretical foundations are based on traditional linear physics, there is no written work yet that combines them into a unique comprehensive framework, and fully exposes why and how MC implementations of coupled heat transfers are now possible.

With this framework, the full power of the method is then made available both for computations and analysis. In the following, the idea of thermal paths is built up progressively using different formal propositions, each of which is translated into a corresponding algorithm. In the end, the physical images generated by our proposition should allow physicists and engineers to renew their interpretation of the coupling of heat transfers in a given system. It may also open didactic perspectives that will be discussed in forthcoming articles in educational sciences.

## 1.2 Related works

Well beyond the work on kinetic models, there has long been an explicit intention to give a probabilistic interpretation to the linear differential equations of physics (including, in particular, those relating to heat transfer in material media). As a precursor, the famous papers by Courant, Friedrichs and Lewy published in 1928 (translated into English and republished in

1967) [17, 18] sketched the first ideas of a statistical calculation of physical quantities by bringing together random walks and boundary value problem of elliptic equations. A major advance in this field was made by Feynman and Kac when they formally wrote the solutions of a class of parabolic differential equations as an expectation taken over stochastic processes. This immediately opened up the possibility of treating a set of linear problems initially formulated in deterministic terms, through the sampling of Brownian path space [19, 20]. This was followed by a large number of applications in different fields of physics associated with constant and regular developments of Monte Carlo methods for solving linear differential equations; the interested reader can explore the citation trees starting, for example, from [21–26]. Connections are direct with the literature of Monte Carlo methods for solving linear algebraic systems when space-time discretization schemes are used as numerical approximations (see for example the seminal work of [27]).

Regarding the diffusion equation (which is the usual representation of energy conduction phenomena), the random Walk on Sphere (WoS) algorithm introduced in [28] was immediately perceived as a conceptual breakthrough due to its astonishing convergence properties and became quickly a very popular method. For a Brownian process, its principle is to sample a first passage point on the largest sphere contained in the bounded domain (centered around the considered point) by probabilizing the Green's function of the spatio-temporal diffusion operator. From this principle, many nice works were born based on comparable ideas to deal with situations with heterogeneous diffusion coefficients, or to improve near-wall approximations, or even to extend to other types of elementary primitives [29–36].

More specifically focused on applications to heat conduction phenomena, Hji-Sheikh and Sparrow [37] popularized the concept of floating random walk [38], meaning that it is possible to get rid of spatial grids previously introduced as a support for numerical resolution (meshless methods). This work has rapidly percolated into the field of numerical computation (see for example [39, 40]) from which many improvements have been developed such as the possibility to treat anisotropic media [41], to take into account all types of boundary conditions (Robin or Newman conditions) [42] or to include spatial dependencies on the conductivity tensor and the source field [43]. For a more exhaustive entry in the subject, the last editions of Ozisik's book compile a state of the art of Monte Carlo methods for thermal conduction [44]. Echoing quite naturally the applicative needs for tackling heat transfer in fluids, several works have extended the previous proposals to cases where energy transfers are no longer only conductive but result from advecto-diffusive phenomena [45–47]. Alternative approaches led to somewhat different Monte Carlo algorithms for heat conduction based on the linear Boltzmann equation, which is known to give back the diffusive phenomenology in the limit of low Knudsen numbers [48, 49].

Given the well-known ability of Monte Carlo methods to deal efficiently with complex geometries, different authors have proposed to use this statistical approach to tackle thermal conduction problems when geometrical refinements of the supporting scenes make the usual numerical methods difficult to implement, if not merely unusable [50–52]. However, when it comes to extending the idea to coupled thermal phenomena, there are no statistical proposals in the literature that manage to retain both the flexibility and the decisive scaling properties. Some works in the field of conducto-radiative coupling are quite relevant and allow considerable conceptual advances [53–57]—but they are somewhat distant from adjoint Monte Carlo methods.

To our knowledge, apart from the contributions that allowed the development of the present framework, no coupled conducto-convecto-radiative problem treated in a single Monte Carlo algorithm in realistic geometry has been proposed so far.

## 1.3 The theoretical framework

In this framework, the objective is to estimate a given quantity at so-called "probe points" such as the temperature at a given point in space at a given time. Such probes can be as well quantities integrated over time, surface and volume, in which case the integral is estimated statistically, without solving the detailed integrand. It is therefore fundamentally different from the "swarm" MC algorithms [58] or "direct" algorithms [59] that aim to evaluate the whole field but are limited in their capacity to tackle high levels of geometrical, temporal or phenomenological complexity.

The main question is then to express the temperature at any location as the expectation of a random variable in the pure filiation of the MC method as it is used in linear transport physics. Indeed, expressing this quantity as a definite integral, which can be considered as a unique statistical estimator, is the guarantee that all the good properties of the method are preserved. This practice of the MC method presents the advantage of preserving an intuitive understanding by analogy with the underlying physics, while systematically providing the associated formalisation (for foundational references, see [60, 61]).

In Section 3 "Linearity and Propagators", we show how to build this expectation for a particular physical study case, using propagation formalism [62–65]. The objective is to express the quantity of interest at the probe point, as an integral over the boundary conditions (here "boundary conditions" comprise the edge conditions and the initial condition) and the sources in the field, using a propagator: the role of the propagator is to quantify the relative contributions by the boundary conditions and by the sources. Here, the corresponding formulations are backward formulations and are generally built using adjoint-based methods. These integral formulations are then reformulated as expectations in order to design MC algorithms. At this stage, the generality of the approach, which simply reflects the essential property of linear physics, should be distinguished from the ability to produce explicit calculations (in the analytical sense), most often limited to academic configurations where the propagator is explicitly known [64, 66–73]. Many works have proposed quite useful and interesting extensions, some of them in terms of applicability domains [34–36, 38, 74–79], but, as far as sampling is concerned, addressing problems with high geometrical complexity remains either very cumbersome or impractical.

To circumvent these difficulties, we turn to the theory of Stochastic Processes [80]. Building upon the work of Feynman and Kac on the statistical representations of parabolic equations [19, 20, 81, 82], we show how the temperature at the probe point can also be expressed as an expectation over a stochastic process. Each realization of the process is a path that starts at the probe point and ends at a boundary condition or at a source, thereby sampling the various possible contributions to the quantity of interest. Averaging a large number of contributions sampled from either the propagator or the stochastic process yields an unbiased estimate of the same expectation, that is, both methods converge to the very same value, although by sampling different path-spaces. This strict equivalence means that stochastic processes can be used instead of propagators, yielding two important advantages: first, stochastic processes can be sampled efficiently even in the presence of complex geometry, and second, since different procedures can be designed to sample a given stochastic process, algorithms can be optimized.

In the part of MC literature that discusses Green's formalism or stochastic processes, the question of coupling phenomenologies of various physics is not an issue that is addressed as such, and this is for very different reasons in each case. On the one hand, in Green's formalism, coupling by sources is self-evident due to the formalism itself, and there are no conceptual difficulties associated to this question. Yet, from an implementation point of view, if MC proposals are only based on this formalism, they are essentially limited to academic cases and do not

fit in the present framework. On the other hand, in the stochastic processes formalism, the path description conveys an intuitive picture by analogy with corpuscular tracking, but this picture remains narrowly limited to the phenomenology that is being treated. In this case, the coupling between different phenomenologies is an issue that has no self-evident formal translation. As a consequence, proposals of MC probe point algorithms for solving coupled thermal transfers are scarce in the literature.

Any theoretical framework that claims to deal with coupled thermal transfers by MC must articulate how paths should be sampled at the coupling locations. We make an extensive use of the double randomization principle [83, 84] as it allows the sampling procedure to rely on a local description of the probabilistic model at each step, including at the coupling locations, whereas the integral approach of Feynman-Kac is built upon a global understanding of the physical subsystems and is hence much more challenging to implement.

## 1.4 The particular model supporting the presentation of the theoretical development

A particular heat transfer model has been chosen to support the exposition of the framework. It is described in Section 2 and summarized in Eq (6). This choice of model is by no means limiting, and various other choices could have been made, such as those described in [10, 14].

Each fundamental element of the proposal will be described in a general manner, and its translation and consequences illustrated in this specific case. In particular, all the choices that aim at preserving the good properties of the MC method, that is, its inherent power of analysis and computational practicability (insensitivity to geometrical refinements, ease of implementation. . .) will be emphasized. This practicability is allowed by the tools developed for image synthesis: ray-tracing and grid acceleration, such that the computation time is almost insensitive to the degree of refinement of geometrical data. As it is crucial for us to preserve the ability of the MC method to scale up to infinite geometric complexity, we ensure that our algorithms are compatible with the well-established tools developed in the computer graphics community. This implies that the interaction between the algorithms and the geometry must essentially rely on scanning the scene through state-of-the-art ray-surface intersectors [85–87]. To ensure both compatibility with ray-surface intersectors and flexibility of implementation, we show in the last part that the practical implementation of MC algorithms on any geometry requires to formulate an approximation of the description of Brownian motions. The proposed approximation is theoretically justified and validated on different application cases. Notwithstanding, it could easily be replaced or modified in case of specific needs.

## 2 The thermal model

The thermal model used below for illustrative purposes is fully described in the present section.

Throughout the text, the symbol $\theta$ is reserved to designate temperatures (with notably extended meanings for the radiative model). For all other symbols, we give a full nomenclature in S1 File.

The conductive and convective transfers are linear and the radiative transfer is linearized around a reference temperature $\theta_{\mathrm{ref}}$. The thermophysical parameters are (possibly constant) functions of space. The geometry is arbitrary and the spatial representations are three-dimensional. If necessary, the formalism also allows to reduce the descriptive dimension according to the symmetries of the system.

These choices are motivated by our intention of proposing a linear heat transfer model that gathers most of the conceptual difficulties with respect to the objective of the paper: demonstrating how the coupling between the transfer modes is carried out in a probabilized

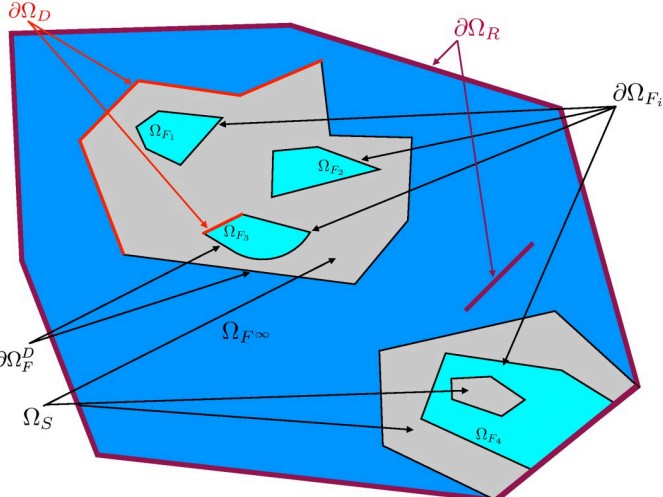

**Fig 1. Illustration of a conducto-convecto-radiative configuration.** The solid domain $\Omega_S$ is shown in gray, $m$ fluid cavities $\Omega_{F_i}$ are shown in light blue, and the surrounding fluid cavity $\Omega_{F^\infty}$ is shown in dark blue. Radiation is present within the whole scene and the system is entirely semi-transparent. Conduction takes place only in solids.

formalism. Of course, this could have been done with somewhat different models (especially for the convective and radiative parts). For example, here, radiation takes place in a semi-transparent medium rather than exchanging between opaque surfaces; the convective cavities are described with an unsteady model and perfectly homogeneous temperature, rather than by an advecto-diffusive description of the temperature; Neumann-type boundary conditions (heat flux imposed at the boundary), as well as situations with volumic power sources only, are left aside (their consideration is a bit different but is not a source of difficulty).

As an example, Fig 1 presents a geometrical abstraction of a configuration with solid subsystems and fluid cavities where the three types of heat transfers can operate in a coupled manner. This type of configuration, associated with the chosen model, depicts a whole diversity of situations encountered in physical or technological questions.

## 2.1 Radiative heat transfer

From the point of view of radiative transfer, the system is entirely semi-transparent (both in the solid and fluid parts). The radiative transfer equation that governs this phenomenology allows to take into account the effect of absorption and scattering of radiation within the system, even where optical properties are heterogeneous.

The radiative transfer equation (RTE) is given with the following two restrictions:

- The refractive index is uniform throughout the system (no ray curvature within solids, no refraction/reflection at solid/fluid interfaces). The writings are simplified and none of our conclusions would be impeded by alleviating this assumption.

- The equation is linearized around the specific equilibrium intensity at a temperature $\theta_{\text{ref}}$. This assumption is fundamental to remain in the framework of linear heat transfers, and it is widely used in applications where the temperature differences in the system are small compared to the absolute temperatures.

Under these assumptions, the radiative model is given by the usual radiative transfer equation written in monochromatic specific intensity $I_\nu \equiv I_\nu(\vec{x}, \vec{u}, t)$ at position $\vec{x}$, in direction $\vec{u}$ at time $t$ and at frequency $\nu$ (Eq (64) in Appendix A).

To ensure formal coherence for the coupling with the other transfer modes, we replace the specific intensity by its translation in terms of radiance temperature $\theta_{R,\vec{u}}^{v}$ (also known as brightness temperature, which is widely used in the experimental field). Appendix A presents how the version written in monochromatic radiance temperature in the $\vec{u}$ direction reads:

$$\vec{u}.\vec{\nabla}\theta_{R,\vec{u}}^{v} = -k_{e}^{v}\theta_{R,\vec{u}}^{v} + k_{a}^{v}\theta + k_{s}^{v}\int_{\mathbb{S}^{2}}p_{S}^{v}(\vec{u}|\vec{u}')\mathrm{d}\vec{u}'\theta_{R,\vec{u}'}^{v} \tag{1}$$

where $k_{a}^{v}, k_{s}^{v}, k_{e}^{v} = k_{a}^{v} + k_{s}^{v}$ are the monochromatic absorption, scattering and extinction coefficients, $p_{S}^{v}$ is the scattering phase function, and $\mathbb{S}^{2}$ denotes the unit sphere in a three dimension space.

It should be noted that even if the radiative equation is taken to be stationary, the radiance temperature $\theta_{R,\vec{u}}^{v}(\vec{x}, t)$ depends on time because $\theta$ evolves in the system due to the coupling with conductive and convective processes.

The energy conservation equations for solid and fluid parts (heat equation for a semi-transparent medium) are formulated below in such a way that the radiative contribution appears through the radiative power density $\psi_{R} \equiv \psi_{R}(\vec{x}, t)$ defined as the difference between the absorbed and the emitted power densities. Under the stated assumptions, it can be written in the following form (see Appendix A):

$$\psi_{R} = \zeta \int_{0}^{+\infty} p_{N}(v)\mathrm{d}v \int_{\mathbb{S}^{2}}\frac{1}{4\pi}\mathrm{d}\vec{u}\left(\theta_{R,\vec{u}}^{v} - \theta\right) \tag{2}$$

where $\zeta = 16k_{a}\sigma\theta_{\mathrm{ref}}^{3}$ is the linearized radiative transfer coefficient and $p_{N}(v)$ is defined by Eq (75).

The complete formulation of the radiative model then reads:

$$\begin{cases} \psi_{R} &= \zeta(\theta_{R} - \theta) \\[2mm] \theta_{R} &= \int_{\mathbb{S}^{2}}\frac{1}{4\pi}\mathrm{d}\vec{u} \ \theta_{R,\vec{u}} \\[2mm] \vec{u}.\vec{\nabla}\theta_{R,\vec{u}} &= -k_{e}\theta_{R,\vec{u}} + k_{a}\theta + k_{s}\int_{\mathbb{S}^{2}}p_{S}(\vec{u}|\vec{u}')\mathrm{d}\vec{u}'\theta_{R,\vec{u}'} \end{cases} \tag{3}$$

The temperature $\theta_{R}$ is often called the radiative temperature (the angular integral of the radiance temperature).

To close the radiative problem, an incident radiance temperature $\theta_{R,\partial\Omega_{R},\vec{u}}$ is imposed at a fictitious boundary $\partial\Omega_{R}$ enclosing the whole domain, with $\vec{u}$ entering the domain ($\vec{u} \in \mathbb{S}_{+}^{2}$).

## 2.2 Heat equation for solid sub-domain

The thermal model for the solid is established by combining diffusive energy transfer with the radiative source term described above. The conductive energy flux density vector is classically given by the Fourier law $\vec{j} = -\lambda\vec{\nabla}\theta_{S}$ in which $\lambda$ is the thermal conductivity of the material and $\theta_{S} \equiv \theta_{S}(\vec{x}, t)$ is the local temperature of the solid.

Thus, the local energy balance in the semi-transparent solid is written as follows:

$$\rho C\partial_{t}\theta_{S} = \underbrace{-\vec{\nabla}.(-\lambda\vec{\nabla}\theta_{S})}_{\text{conductive exchange}} + \underbrace{\zeta(\theta_{R} - \theta_{S})}_{\text{radiative exchange}} \tag{4}$$

where $\rho$ and $C$ are the mass density and the heat capacity of the material. $\zeta$ and $\theta_R$ were defined in the previous section.

As illustrated in Fig 1, the material medium under consideration is not necessarily connected but is bounded by a surface denoted $\partial\Omega_S$ where two types of conditions can be imposed (Neumann-type boundary conditions are not considered in this text):

- Dirichlet-type boundary conditions: $\theta_S = \theta_D$ on the interface $\partial\Omega_D$ where $\theta_D$ is the imposed temperature,

- Robin-type boundary conditions: $-\lambda\vec{n}.\vec{\nabla}\theta_S = -\lambda\frac{\partial\theta_S}{\partial n} = h_F(\theta_F - \theta_S)$ on the complementary surface to $\partial\Omega_D$ noted $\partial\Omega_S\backslash\partial\Omega_D$.
  $\vec{n}$ is the incoming normal to the solid surface, $\theta_F$ is the temperature of the fluid whose model will be given in the following section and $h_F$ is the local convective exchange coefficient. Robin's condition specifies the coupling constraint between the solid and the fluid by simply stating the continuity of the flux at the solid-fluid interface.

The initial condition at time $t_I$ is generically noted $\theta_I \equiv \theta_I(\vec{x})$.

## 2.3 Heat equation for fluid sub-domain

As illustrated in Fig 1, the fluid domain is composed of $m$ cavities noted $\Omega_{F_i}(i \in \{1, 2, \ldots, m\})$ and of a domain $\Omega_{F\infty}$ partially or totally enclosing the $\Omega_S$ solid domain. The fluid domain is therefore the union of these $m + 1$ domains:

$$\Omega_F \equiv \Omega_{F_1} \cup \Omega_{F_2} \cup \cdots \cup \Omega_{F_m} \cup \Omega_{F\infty}$$

In each cavity $\Omega_{F_i}$ (of volume $\mathcal{V}_{F_i}$) and each surface $\partial\Omega_{F_i}$, the fluid is assumed to be perfectly mixed by convection at each instant and the heat flux density at the solid walls is modeled by a linear transfer law (Newton's law). Under this assumption, the temperature of the fluid contained in the i$^{\text{th}}$ cavity, noted $\theta_{F_i}$ is spatially uniform and varies only as a function of time. The mass density $\rho_i$, the heat capacity $C_i$ and the linearized radiative heat transfer coefficient $\zeta_i$ are also spatially uniform.

The temperature of the fluid contained in the enclosing cavity $\Omega_{F\infty}$ of surface $\partial\Omega_{F\infty}$ is imposed, it is noted $\theta_{F\infty}$ and can depend on time.

The energy balance of the semi-transparent fluid inside the i$^{\text{th}}$ cavity is as follows:

$$\rho_i C_i \mathcal{V}_{F_i}\frac{\mathrm{d}\theta_{F_i}}{\mathrm{d}t}(t) = \underbrace{\zeta_i\int_{\Omega_{F_i}}(\theta_R(\vec{x}_R, t) - \theta_{F_i}(t))\mathrm{d}\vec{x}_R}_{\text{radiative exchange}} + \underbrace{\int_{\partial\Omega_{F_i}}h_F(\vec{y}_s)(\theta_s(\vec{y}_s, t) - \theta_{F_i}(t))\mathrm{d}\vec{y}_s}_{\text{convective exchange}} \quad (5)$$

$\zeta$ and $\theta_R$ are defined in the section on radiative transfer. The initial condition at time $t_I$ is $\theta_I$.

In this correlative model, which is widely used in industrial thermal issues, the convective exchange coefficient $h_F$ summarizes the phenomenological complexity of the exchanges between the fluid and the wall (here $h_F$ depends explicitly on the position at the wall; it may also depend on time with no additional difficulties). Many works in the literature aim at providing correlations adapted to different study cases, generally parameterized with a set of dimensionless numbers [88, 89]. As mentioned above, diffusion-drift models for temperature could have been chosen without changing our main message. As an example, the work described in [14] is focused on the MC resolution of convective-conducting-radiative coupled models using a local heat equation for the fluid (with a prescribed velocity field).

## 2.4 Summary: The coupled model

Let us define, for any domain $\Omega_J$: the adherence $\bar{\Omega}_J = \Omega_J \bigcup \partial\Omega_J$ and the interior $\mathring{\Omega}_J = \Omega_J \setminus \partial\Omega_J$. For boundary surfaces, the complementary of $\partial\Omega_J$ to $\partial\Omega_K$ is denoted $\partial\Omega_J^K = \partial\Omega_J \setminus \partial\Omega_K$.

As illustrated in Fig 1, the domain consists of:

- an enclosing fluid cavity $\Omega_{F\infty}$ of boundary $\partial\Omega_{F\infty}$, in which the temperature $\theta_{F\infty}$ is known;

- $m$ fluid cavities noted $\Omega_{F_i}$ of boundary $\partial\Omega_{F_i}$;

- a solid domain $\Omega_S$ of boundary $\partial\Omega_S = \partial\Omega_D \cup \partial\Omega_S^D$ with a Dirichlet condition on $\partial\Omega_D$ and a Robin condition on $\partial\Omega_S^D = \partial\Omega_{F\infty}^D \cup_{i=1}^m \partial\Omega_{F_i}^D$;

- a fictitious boundary to handle the radiative boundary condition $\partial\Omega_R$.

Let $\theta$ denote the spatio-temporal temperature field, solution of the following system:
a) Solid & Fluid

$$\begin{cases} \theta \equiv \theta_S, & \vec{x} \in \bar{\Omega}_S, & t \in ]t_I, +\infty[ \\ \theta \equiv \theta_{F_i}, & \vec{x} \in \mathring{\Omega}_{F_i}, & t \in ]t_I, +\infty[ \\ \theta \equiv \theta_{F\infty}, & \vec{x} \in \mathring{\Omega}_{F\infty}, & t \in ]t_I, +\infty[ \end{cases} \qquad (6a)$$

b) Solid, $i \in \{1, \ldots, m\}$

$$\begin{cases} \rho C \partial_t \theta_S = -\vec{\nabla}.(-\lambda\vec{\nabla}\theta_S) + \zeta(\theta_R - \theta_S) & , \vec{x} \in \mathring{\Omega}_S, & t \in ]t_I, +\infty[ \\ \theta_S = \theta_I & , \vec{x} \in \bar{\Omega}_S, & t = t_I \\ \theta_S = \theta_D & , \vec{y} \in \partial\Omega_D, & t \in ]t_I, +\infty[ \\ -\lambda\dfrac{\partial\theta_S}{\partial n} = h_F\left(\theta_{F_i} - \theta_S\right) & , \vec{y} \in \partial\Omega_{F_i}^D, & t \in ]t_I, +\infty[ \\ -\lambda\dfrac{\partial\theta_S}{\partial n} = h_F(\theta_{F\infty} - \theta_S) & , \vec{y} \in \partial\Omega_{F\infty}^D, & t \in ]t_I, +\infty[ \end{cases} \qquad (6b)$$

c) Fluid, $i \in \{1, \ldots, m\}$

$$\begin{cases} \rho_i C_i \mathcal{V}_{F_i} \dfrac{d\theta_{F_i}}{dt} = \zeta_i \displaystyle\int_{\Omega_{F_i}} (\theta_R(\vec{x}_R, t) - \theta_{F_i}(t))d\vec{x}_R \\ \qquad\qquad + \displaystyle\int_{\partial\Omega_{F_i} \cap \partial\Omega_S} h_F(\vec{y}_S)(\theta_S(\vec{y}_S, t) - \theta_{F_i}(t))d\vec{y}_S \\ \qquad\qquad + \displaystyle\int_{\partial\Omega_{F_i} \cap \partial\Omega_D} h_F(\vec{y}_S)(\theta_D(\vec{y}_S, t) - \theta_{F_i}(t))d\vec{y}_S, \vec{x} \in \mathring{\Omega}_{F_i}, t \in ]t_I, +\infty[ \\ \qquad \theta_{F_i} = \theta_I, \vec{x} \in \mathring{\Omega}_{F_i} & , t = t_I \end{cases} \qquad (6c)$$

d) Solid & Fluid

$$\begin{cases} \theta_R(\vec{x}, t) &= \displaystyle\int_{\mathbb{S}^2} \frac{1}{4\pi} d\vec{u}\, \theta_{R,\vec{u}}(\vec{x}, t) & , \vec{x} \in \overset{\circ}{\Omega},\ t \in ]t_I, +\infty[ \\[2ex] \vec{u}.\vec{\nabla}\theta_{R,\vec{u}} &= -k_e \theta_{R,\vec{u}} + k_a \theta + k_s \displaystyle\int_{\mathbb{S}^2} p_S(\vec{u}|\vec{u}') d\vec{u}' \theta_{R,\vec{u}'} & , \vec{x} \in \overset{\circ}{\Omega}, \vec{u} \in \mathbb{S}^2,\ t \in ]t_I, +\infty[ \\[2ex] \theta_{R,\vec{u}} &= \theta_{R,\partial\Omega_R,\vec{u}} & , \vec{y} \in \partial\Omega_R, \vec{u} \in \mathbb{S}^2_+,\ t \in ]t_I, +\infty[ \end{cases} \quad (6d)$$

In the following sections, we present the two fundamental proposals that will permit:

- to express the temperature $\theta(\vec{x}, t)$ at position $\vec{x}$ at time $t$, solution of System (6), as the expectation of a random variable. This first proposal proceeds from the rewriting of the problem in Green's formalism. The theory will first be given in full generality (Section 3), then applied for each of the three heat transfer modes separately (Section 4), and finally to the coupled model (Section 5);

- to define the thermal paths that ensure that this random variable can be sampled by a MC method. This second proposal introduces the notion of trajectory from the theory of stochastic processes and yields the construction of a path space for sampling. The theory and its application to the thermal model under consideration are first provided (Section 6), and then a practical proposition for sampling the conductive paths is developed (Section 7).

## 3 Linearity and propagators

The objective of this section is to produce a probabilization of the expressions in order to write the quantity of interest as the expectation of a random variable. The link with the MC method is made explicite by pseudo-algorithms that describe the sampling procedure of this random variable.

### 3.1 The formal proposition

The thermal sub-models described in the previous section by Eqs (3), (4) and (5), along with their respective boundary conditions, are all linear and can be formally written in a similar way. This leads to a system of linear integro-differential equations representing a well-posed boundary value problem, which can be written generically in the following operational form:

$$\begin{cases} c(\vec{w}, t)\partial_t f(\vec{w}, t) + L(f)(\vec{w}, t) = a(\vec{w}, t) f_{\mathcal{W}}(\vec{w}, t) & , t \in [t_I, +\infty[ & , \vec{w} \in \overset{\circ}{\mathcal{W}} \\[1.5ex] L_{\partial\mathcal{W}}(f)(\vec{w}, t) = f_{\partial\mathcal{W}}(\vec{w}, t) & , t \in ]t_I, +\infty[ & , \vec{w} \in \partial\mathcal{W} \\[1.5ex] f(\vec{w}, t_I) = f_I(\vec{w}) & & , \vec{w} \in \bar{\mathcal{W}} \end{cases} \quad (7)$$

where the temporal dimension of the problem is specifically marked and the corresponding variable is noted $t$. The non-temporal part of the integration domain on which the model is built can be of any dimension and noted $\mathcal{W}$; the vector $w$ represents a way to name a point in this space. Depending on the model, $\mathcal{W}$ may simply be the geometric space, as in the model for solids, or the space of locations and directions (phase space) in the case of radiative transfer. The particular case where this space has dimension zero can be treated without difficulty and is not the object of a specific development; it is typically the case for the fluid model. $L$ and $L_{\partial\mathcal{W}}$ are homogeneous and linear integrodifferential operators (here homogeneity means that

there are no constant terms). $f_{\mathcal{W}}$, $f_{\partial\mathcal{W}}$ and $f_I$ are source terms with the same dimension as $f$ (they are assumed to be prescribed functions for the moment).

Functions $c(\vec{w}, t)$ and $a(\vec{w}, t)$ are part of the problem definition and they are known. $f$, $f_{\mathcal{W}}$, $f_{\partial\mathcal{W}}$ and $f_I$ are real-valued functions.

Based on Eq (7), we construct a generic form for the sub-models.

**Restrictions**. We will restrict the proposition to models which satisfy the second principle, or H theorem, as it is understood in thermodynamics. More precisely, we only consider systems that exhibit an equilibrium solution (understood here as the uniformity of the $f$ function) for particular conditions on the sources. In the generic framework of Eq (7), this implies that the model must satisfy the following property:

$$\forall a_1, a_2, a_3 \in \mathbb{R}, f_{\mathcal{W}} = a_1, f_{\partial\mathcal{W}} = a_2, f_I = a_3 \;\Rightarrow\; \min(a_1, a_2, a_3) < f(\vec{w}, t) < \max(a_1, a_2, a_3) \quad (8)$$

The thermal model that has been presented above in System (6) satisfies this property. Similar choices of generic models satisfying this equilibrium property could have been made. To provide a counter-example, if the balance equation for the solid sub-domain was a diffusion equation with a prescribed power source instead of the radiative term $\zeta(\theta_R - \theta_S)$ (for example, the contribution of an electric heater, without any loss) the equilibrium condition Eq (8) can not be satisfied. The same conclusion is obtained with a non-zero imposed flux (Neumann boundary condition). This does not mean that it is not possible to probabilize these models up to a Monte-Carlo implementation, but the strategies to be implemented for this are quite specific and lead to ad-hoc propositions that will be detailed in dedicated papers (some propositions are already implemented in the Stardis library).

## 3.2 Expression for $f(\vec{w}, t)$

Since the model is linear, one can write the solution of Eq (7) as

$$
\begin{aligned}
f(\vec{w}, t) = \;\; &+ \int_{\mathcal{W}} g_I(\vec{w}, t | \vec{w}_I, t_I) f_I(\vec{w}_I) \mathrm{d}\vec{w}_I \\
&+ \int_{\partial\mathcal{W}} \int_{t_I}^{t} g_{\partial\mathcal{W}}(\vec{w}, t | \vec{w}_{\partial\mathcal{W}}, t_{\partial\mathcal{W}}) f_{\partial\mathcal{W}}(\vec{w}_{\partial\mathcal{W}}, t_{\partial\mathcal{W}}) \mathrm{d}t_{\partial\mathcal{W}} \mathrm{d}\vec{w}_{\partial\mathcal{W}} \quad (9) \\
&+ \int_{\mathcal{W}} \int_{t_I}^{t} g_{\mathcal{W}}(\vec{w}, t | \vec{w}_{\mathcal{W}}, t_{\mathcal{W}}) f_{\mathcal{W}}(\vec{w}_{\mathcal{W}}, t_{\mathcal{W}}) \mathrm{d}t_{\mathcal{W}} \mathrm{d}\vec{w}_{\mathcal{W}}
\end{aligned}
$$

where the functions $g_I$, $g_{\partial\mathcal{W}}$ and $g_{\mathcal{W}}$ are the propagators for the different sources (respectively, the initial condition, on the surface, in the volume).

Reading Eq (9) is quite intuitive as it combines the concepts of superposition and causality: the observable $f$ at point $\vec{w}$ and time $t$ results from the effects of three sources in the sense of Green's theory (that is, the inhomogeneous (right-hand) terms in Eq (7)):

- the effect of the initial condition $f_I$ at any point in phase space $\vec{w}_I \in \mathcal{W}$ and at time $t_I$, provided by the propagator $g_I(\vec{w}, t | \vec{w}_I, t_I)$,

- the effect of the boundary conditions $f_{\partial\mathcal{W}}$ at any point on the edge of phase space $\vec{w}_{\partial\mathcal{W}} \in \partial\mathcal{W}$ and at any time $t_{\partial\mathcal{W}} \in [t_I, t]$, provided by the propagator $g_{\partial\mathcal{W}}(\vec{w}, t | \vec{w}_{\partial\mathcal{W}}, t_{\partial\mathcal{W}})$,

- the effect of the source $f_{\mathcal{W}}$ at any point in phase space $\vec{w}_{\mathcal{W}} \in \mathcal{W}$ and at any time $t_{\partial\mathcal{W}} \in [t_I, t]$, provided by the propagator $g_{\mathcal{W}}(\vec{w}, t | \vec{w}_{\mathcal{W}}, t_{\mathcal{W}})$.

### 3.3 The Green function $g(\vec{w}, t | \vec{w}', t')$

In most non-academic systems, it is not possible to obtain the explicit form of the propagators. Nevertheless, propagators are the solution of linear mathematical models which can most often be written without much difficulty. The production of adjoint models, as well as Green's formalism, traditionally provide a unifying technical framework for this purpose.

Hereafter we briefly describe the Green's formalism approach in the case of Eq (7). Technically, the contribution of the sources are constructed from Dirac distributions and convolution operators. Since the equation is linear, the solution $f$ can be reconstructed by superposition (see [62, 90–92] for more details).

From Eq (7), the following system is constructed:

$$\begin{cases} c(\vec{w}, t)\partial_t g(\vec{w}, t | \vec{w}', t') + L(g)(\vec{w}, t | \vec{w}', t') = \delta(\vec{w} - \vec{w}') \qquad \delta(t - t'), \\[2mm] \qquad\qquad\qquad\qquad\qquad\qquad\qquad\qquad\qquad t, t' \in ]t_I, +\infty[, \vec{w} \in \bar{\mathcal{W}}, \vec{w}' \in \mathcal{W} \\[2mm] L_{\partial w}(g)(\vec{w}, t | \vec{w}', t') = 0, \qquad\qquad\qquad t, t' \in ]t_I, +\infty[, \vec{w} \in \bar{\mathcal{W}}, \vec{w}' \in \partial\mathcal{W} \\[2mm] g(\vec{w}, t | \vec{w}', t') = 0, \qquad\qquad\qquad\qquad t < t', \vec{w} \in \mathcal{W}, \vec{w}' \in \bar{\mathcal{W}} \end{cases} \quad (10)$$

where the volume source $f_{\mathcal{W}}$ have been replaced by a Dirac distribution $\delta$ in phase space and time, centered at $(\vec{w}', t')$, and where the boundary and initial conditions have been made homogeneous (there is no source except $f_{\mathcal{W}}$). Intuitively, $g(\vec{w}, t | \vec{w}', t')$ can be considered as the effect of a point source at $(\vec{w}', t') \in \bar{\mathcal{W}} \times \mathbb{R}$ on the quantity of interest $f$ at point $(\vec{w}, t)$. The last line in Eq (10) ensures causality—it simply reflects the idea that an effect cannot occur before its cause—and at the same time, it closes the problem by providing initial conditions.

Then, propagators $g_I, g_{\partial\mathcal{W}}$ and $g_{\mathcal{W}}$ are directly constructed from $g$:

$$g_I(\vec{w}, t | \vec{w}_I, t_I) = c(\vec{w}_I, t_I) g(\vec{w}, t | \vec{w}_I, t_I)$$

$$g_{\mathcal{W}}(\vec{w}, t | \vec{w}_{\mathcal{W}}, t_{\mathcal{W}}) = a(\vec{w}_{\mathcal{W}}, t_{\mathcal{W}}) g(\vec{w}, t | \vec{w}_{\mathcal{W}}, t_{\mathcal{W}})$$

$g_{\partial\mathcal{W}}(\vec{w}, t | \vec{w}_{\partial\mathcal{W}}, t_{\partial\mathcal{W}})$ does not have a generic expression without knowledge of the operator $L_{\partial\mathcal{W}}$; it will be addressed on a case-by-case basis later on.

### 3.4 Probabilistic reformulation

The following quantities are introduced:

$$\begin{aligned} p_I &\equiv p_I(\vec{w}, t | t_I) & &= \int_{\mathcal{W}} g_I(\vec{w}, t | \vec{w}_I, t_I) \mathrm{d}\vec{w}_I \\[2mm] p_{\partial\mathcal{W}} &\equiv p_{\partial\mathcal{W}}(\vec{w}, t | t_I) & &= \int_{\partial\mathcal{W}} \int_{t_I}^{t} g_{\partial\mathcal{W}}(\vec{w}, t | \vec{w}_{\partial\mathcal{W}}, t_{\partial\mathcal{W}}) \mathrm{d}t_{\partial\mathcal{W}} \mathrm{d}\vec{w}_{\partial\mathcal{W}} \\[2mm] p_{\mathcal{W}} &\equiv p_{\mathcal{W}}(\vec{w}, t | t_I) & &= \int_{\mathcal{W}} \int_{t_I}^{t} g_{\mathcal{W}}(\vec{w}, t | \vec{w}_{\mathcal{W}}, t_{\mathcal{W}}) \mathrm{d}t_{\mathcal{W}} \mathrm{d}\vec{w}_{\mathcal{W}} \end{aligned} \quad (11)$$

Using the restriction Eq (8) into Eq (9) leads to:

$$p_I + p_{\partial\mathcal{W}} + p_{\mathcal{W}} = 1 \quad (12)$$

Indeed, Green functions only depend on the linear and homogeneous operator parts in Eq (7) (left-hand side of the equations). This property can be demonstrated by considering any

value for the sources, for example $a_1 = a_2 = a_3$. Eq (12) enables us to consider $p_I$, $p_{\partial \mathcal{W}}$ and $p_{\mathcal{W}}$ as probabilities in the following.

Let us define the following independent random variables (r.v.):

$$\mathcal{B}_1(p), ..., \mathcal{B}_n(p) \qquad \text{are } n \text{ independent Bernoulli r.v. with parameter } p$$

$\vec{W}_I \qquad \text{is a r.v. with distribution } p_{\vec{W}_I}(\vec{w}, t | \vec{w}_I, t_I)$

$(\vec{W}_{\partial \mathcal{W}}, T_{\partial \mathcal{W}}) \qquad \text{is a paired r.v. with distribution } p_{(\vec{w}_{\partial \mathcal{W}}, T_{\partial \mathcal{W}})}(\vec{w}, t | \vec{w}_{\partial \mathcal{W}}, t_{\partial \mathcal{W}})$

$(\vec{W}_{\mathcal{W}}, T_{\mathcal{W}}) \qquad \text{is a paired r.v. with distribution } p_{(\vec{w}_{\mathcal{W}}, T_{\mathcal{W}})}(\vec{w}, t | \vec{w}_{\mathcal{W}}, t_{\mathcal{W}})$

where probability density functions are:

$$
\begin{aligned}
p_{\vec{W}_I}(\vec{w}, t | \vec{w}_I, t_I) &= g_I(\vec{w}, t | \vec{w}_I, t_I) / p_I(\vec{w}, t | t_I) \\
p_{(\vec{w}_{\partial \mathcal{W}}, T_{\partial \mathcal{W}})}(\vec{w}, t | \vec{w}_{\partial \mathcal{W}}, t_{\partial \mathcal{W}}) &= g_{\partial \mathcal{W}}(\vec{w}, t | \vec{w}_{\partial \mathcal{W}}, t_{\partial \mathcal{W}}) / p_{\partial \mathcal{W}}(\vec{w}, t | t_I) \\
p_{(\vec{w}_{\mathcal{W}}, T_{\mathcal{W}})}(\vec{w}, t | \vec{w}_{\mathcal{W}}, t_{\mathcal{W}}) &= g_{\mathcal{W}}(\vec{w}, t | \vec{w}_{\mathcal{W}}, t_{\mathcal{W}}) / p_{\mathcal{W}}(\vec{w}, t | t_I)
\end{aligned}
$$

Eq (9) can then be reformulated to write $f(\vec{x}, t)$ as the expectation of a random variable $F$:

$$f = \mathbb{E}[F] \qquad (13)$$

with

$$F = \mathcal{B}_1(p_I) f_I(\vec{W}_I) + (1 - \mathcal{B}_1(p_I))\{\mathcal{B}_2(p_2) f_{\partial \mathcal{W}}(\vec{W}_{\partial \mathcal{W}}, T_{\partial \mathcal{W}}) + (1 - \mathcal{B}_2(p_2)) f_{\mathcal{W}}(\vec{W}_{\mathcal{W}}, T_{\mathcal{W}})\} \quad (14)$$

where $p_2 = \frac{p_{\partial \mathcal{W}}}{1 - p_I}$.

## 3.5 Monte-Carlo algorithm

Based on the above formulations, it is straightforward to construct the sampling algorithm for $F$: i) sample one of the three types of sources according to the probabilities $p_I$, $p_{\partial \mathcal{W}}$ and $p_{\mathcal{W}}$, ii) sample a location and possibly a time according to the corresponding probability density function and iii) keep the value of the source at this sampled location and time (see Algorithm 1).

The MC algorithm estimating $f = \mathbb{E}[F]$ consists in sampling a set of realizations $\hat{f}$ of $F$ and estimating $f$ as the arithmetic mean of this set.

**Algorithm 1**: Sampling algorithm for the random variable $F$ at phase space position $\vec{w}$ and time $t$. $\hat{f}$ is the corresponding realization of the random variable

```
Sample r₁ uniformly on [0, 1];
if r₁ < pᵢ then
   Sample w⃗ᵢ according to the law of W⃗ᵢ;
   f̂ = fᵢ(w⃗ᵢ);
else
   Sample r₂ uniformly on [0, 1];
   if r₂ < p₂ then
      Sample (w⃗∂𝒲, t∂𝒲) according to the law of (W⃗∂𝒲, T∂𝒲);
      f̂ = f∂𝒲(w⃗∂𝒲, t∂𝒲);
   else
      Sample (w⃗𝒲, t𝒲) according to the law of (W⃗𝒲, T𝒲);
      f̂ = f𝒲(w⃗𝒲, t𝒲);
```

Fig 2 illustrates the proposition of probabilization in the simple case where the problem is only time-dependent (there is no integration over $\mathcal{W}$ in this case), as for instance in the fluid sub-domain.

**Mathematical view point**

Let us consider the equation:

$$\begin{cases} \dfrac{df}{dt} = -\alpha\,(f - f^*), & t \in ]t_I, +\infty[ \\ f(t_I) = f_I \end{cases}$$

where $\alpha$ is a constant and $f^* \equiv f^*(t)$.

For $t \in ]t_I, +\infty[, t' \in \mathbb{R}$ we get

$$\frac{1}{\alpha}\frac{dg}{dt}(t|t') + g(t|t') = \delta(t - t')$$

hence

$$\begin{cases} g(t|t') = \mathcal{H}\left(t \geqslant t'\right)\alpha \exp\left(-\alpha(t - t')\right) \\ g_I(t|t') = g(t|t')/\alpha \\ g_\mathcal{W}(t|t') = g(t|t') \end{cases}$$

and

$$f(t) = g_I(t|t_I)f_I + \int_{t_I}^{t} g_\mathcal{W}(t|t')f^*(t')\mathrm{d}t'$$

**Probabilistic view point**

Define the probabilities

$$\begin{aligned} p_I &= g_I(t|t_I) \\ p_\mathcal{W} &= 1 - p_I \\ p_{T_\mathcal{W}}(t|t_\mathcal{W}) &= g(t|t_\mathcal{W})/p_\mathcal{W} \end{aligned}$$

which leads to

$$\begin{aligned} F(t) &= \mathcal{B}(p_I)f_I + (1 - \mathcal{B}(p_I))\,f^*(T_\mathcal{W}) \\ f(t) &= \mathbb{E}[F(t)] \end{aligned}$$

where $\mathcal{B}$ is a Bernoulli r.v. with parameter $p_I$ and $T_\mathcal{W}$ is a r.v. with distribution $p_{T_\mathcal{W}}$.

---

**Algorithm:** Sampling algorithm for $F(t)$

---

Sample $r$ uniformaly on $[0, 1]$;
**if** $r < p_I(t|t_I)$ **then**
 $\hat{f} = f_I$;
**else**
 Sample $t_\mathcal{W}$ according to the law of $T_\mathcal{W}$;
 $\hat{f} = f^*(t_\mathcal{W})$;

**Fig 2. Implementation example for the probabilization proposition in the case of a problem that is only time-dependent.**

## 4 Implementation of the uncoupled thermal model

The previous section has exposed our probabilization strategy based on the propagative formulation of the solutions of linear models when the equilibrium condition Eq (8) is met. We propose here to directly apply the procedure on each submodel in Eq (6), considered independently (*i.e* decoupled from each other). In this section, we will therefore consider that, for each submodel, crossed variables ensuring the coupling are prescribed (for instance, if we consider a fluid cavity, the solid temperature $\theta_S$ at the interface is assumed to be known).

### 4.1 Fluid sub-domain

Eq (6c) details the contributions on the fluid sub-domain boundary $\partial\Omega_{F_i}$ so that the coupling can be expressed unequivocally. But the distinction between the different parts of the boundary ($\partial\Omega_{F_i} \cap \partial\Omega_D$ and $\partial\Omega_{F_i} \cap \partial\Omega_S$) is useless here as we aim to first formalize the uncoupled problem by assuming that the temperature is known on all of them. Thus, we just start from the generic form provided by Eq (5) in which we write the development as if temperatures $\theta_R$ and $\theta_S$ were known and prescribed time-space functions.

This balance equation and the corresponding probabilization can be written in exactly the same way as in Fig 2. To obtain the final propagative form, there is however an additional step due to the fact that the source term of the differential equation involves integral terms. Appendix B describes these developments that finally lead to the expression of the fluid temperature:

$$\begin{aligned} \theta_{F_i}(t) = &\ g_{F_i,I}(t|t_I)\theta_I \\ &+ \int_{t_I}^{t}\int_{\partial\Omega_{F_i}} g_{F_i,S}(t|\vec{\boldsymbol{y}}_S, \tau)\theta_S(\vec{\boldsymbol{y}}_S, t)\mathrm{d}\vec{\boldsymbol{y}}_S\mathrm{d}\tau \\ &+ \int_{t_I}^{t}\int_{\Omega_{F_i}} g_{F_i,R}(t|\vec{\boldsymbol{x}}_R, \tau)\theta_R(\vec{\boldsymbol{x}}_R, \tau)d\vec{\boldsymbol{x}}_R d\tau \end{aligned} \tag{15}$$

where $g_{Fi}$, $I$, $g_{Fi}$, $S$ and $g_{Fi}$, $R$ stand for the propagation to fluid subvolume $\Omega_{F_i}$ from initial conditions, surface (Solid) and volume (Radiation) respectively.

A random variable whose expectation is the temperature $\theta_{F_i}(t)$ is constructed following the proposition stated in the general case (Eq (7) to Eq (14)), applied to the particular case where the state variable is only time-dependant (as in Fig 2):

$$\theta_{F_i}(t) = \mathbb{E}[\Theta_{F_i}(t)] \tag{16}$$

with

$$\begin{aligned}
\Theta_{F_i}(t) = \quad & \mathcal{B}_1(p_I^{F_i})\theta_I \\
& + (1 - \mathcal{B}_1(p_I^{F_i}))\{\mathcal{B}_2(p_R^{F_i})\theta_R(\vec{X}_R^{F_i}, T^{F_i}) + (1 - \mathcal{B}_2(p_R^{F_i}))\theta_S(\vec{Y}_S^{F_i}, T^{F_i})\}
\end{aligned} \tag{17}$$

To simplify the presentation, the complete definition of the random variables and probabilities involved in this equation are reported to Appendix B.

Algorithm 2 describes the sampling procedure for $\Theta_{F_i}$ defined as above, and Fig 3 illustrates corresponding typical realizations.

**Algorithm 2**: Sampling algorithm for the random variable $\Theta_{F_i}$ defined by Eq (17) assuming the functions $\theta_R$ and $\theta_S$ are known. $\hat{\theta}_{F_i}$ is the corresponding realization of the random variable.

```
Sample r₁ uniformly on [0, 1];
if r₁ < pᵢ^Fᵢ then
    θ̂_Fᵢ = θ_I;
else
    Sample r₂ uniformly on [0, 1];
    if r₂ < p_R^Fᵢ then
        Sample (x⃗_R, τ) according to the law of (X⃗_R^Fᵢ, T^Fᵢ);
        θ̂_Fᵢ = θ_R(x⃗_R, τ);
    else
        Sample (y⃗_S, τ) according to the law of (Y⃗_S^Fᵢ, T^Fᵢ);
        θ̂_Fᵢ = θ_S(y⃗_S, τ);
```

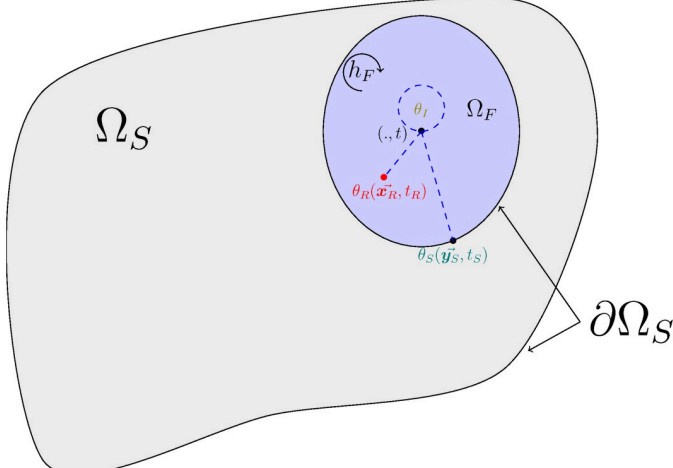

**Fig 3. Illustration of the three possible realizations for $\Theta_{F_i}$.** Each realization represents one of the three contributions that can be returned: initial temperature $\theta_I$, a radiance temperature $\theta_R$ and a boundary temperature with the solid $\theta_S$. The notation $(., t)$ means that the temperature of the fluid is not dependent on the location in the cavity and that the probe can be positioned anywhere.

## 4.2 Solid sub-domain

For the solid sub-domain, we start from the model as described by Eq (6b). In the description of the boundary conditions, temperatures $\theta_D$ and $\theta_{F\infty}$ are known functions of time and space. By contrast, temperatures $\theta_{F_i}$ result from the whole coupling dynamics. As above, we consider that, here, they are prescribed and known.

In agreement with the general form of Eqs (7) and (10), the Green function $g_S \equiv g_S(\vec{x}, t | \vec{x}', t')$ associated with Eq (6b) is solution of:

$$
\begin{cases}
\rho C \partial_t g_S - \vec{\nabla}.(\lambda \vec{\nabla} g_S) + \zeta g_S = \delta(\vec{x} - \vec{x}')\delta(t - t') & , t \in ]t_I, +\infty[, \vec{x} \in \overset{\circ}{\Omega}_S \\[2mm]
g_S = 0 & , t \in ]t_I, +\infty[, \vec{x} \in \partial\Omega_D \\[2mm]
-\dfrac{\lambda}{h_F}\dfrac{\partial g_S}{\partial n} + g_S = 0 & , t \in ]t_I, +\infty[, \vec{x} \in \partial\Omega_F^D \\[2mm]
g_S = 0 & , t < t'
\end{cases}
\tag{18}
$$

The solid temperature can thus be written as:

$$
\begin{aligned}
\theta_S(\vec{x}, t) = \; & \int_{\Omega_S} g_{S,I}(\vec{x}, t | \vec{x}_I, t_I)\theta_I(\vec{x}_I)\mathrm{d}\vec{x}_I \\[2mm]
& + \int_{t_I}^{t} \int_{\partial\Omega_F^D} g_{S,\partial\Omega_F^D}(\vec{x}, t | \vec{y}_F, t_F)\theta_F(\vec{y}_F, t_F)\mathrm{d}\vec{y}_F\mathrm{d}t_F \\[2mm]
& + \int_{t_I}^{t} \int_{\partial\Omega_D} g_{S,\partial\Omega_D}(\vec{x}, t | \vec{y}_D, t_D)\theta_D(\vec{y}_D, t_D)\mathrm{d}\vec{y}_D\mathrm{d}t_D \\[2mm]
& + \int_{t_I}^{t} \int_{\Omega_S} g_{S,R}(\vec{x}, t | \vec{x}_R, t_R)\theta_R(\vec{x}_R, t_R)\mathrm{d}\vec{x}_R\mathrm{d}t_R
\end{aligned}
\tag{19}
$$

where:

- $g_{S,I} = \rho C g_S$ denote propagation from Initial condition,

- $g_{S,\partial\Omega_F^D} = h_F g_S$ denote propagation from surfaces $\partial\Omega_F^D$ corresponding to a Robin condition. In this case, $\theta_F$ takes the value $\theta_{F_i}$ of domain $i$ to which $\vec{y}_F$ belongs,

- $g_{S,\partial\Omega_D} = \lambda \frac{\partial g_S}{\partial n}$ denote propagation from surfaces $\partial\Omega_D$ which a Dirichlet boundary condition ($\theta_D$),

- $g_{S,R} = \zeta g_S$ denote propagation from volume radiation.

A random variable whose expectation is the temperature $\theta_S(\vec{x}, t)$ is constructed following the proposition stated in the general case (Eqs (7) to (14)), with the domain $\mathcal{W}$ consisting in the geometric space $\Omega_S$. It is thus possible to write:

$$
\theta_S(\vec{x}, t) = \mathbb{E}[\Theta_S(\vec{x}, t)]
\tag{20}
$$

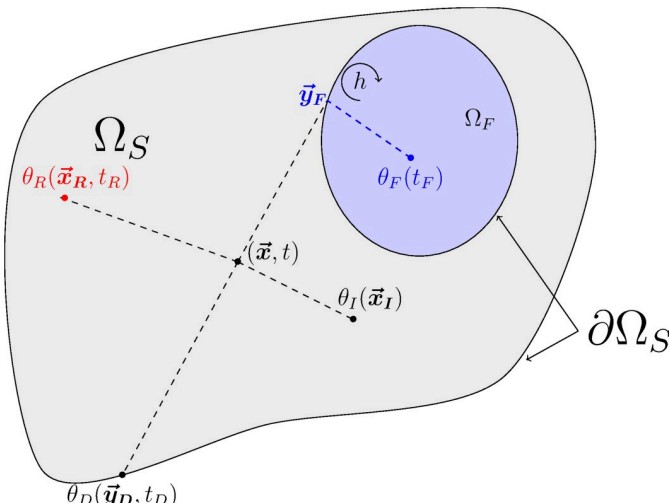

**Fig 4. Illustration of four realizations of $\Theta_S(\vec{x}, t)$):** An initial condition $\theta_I$ at point $\vec{x}_I$, a radiance temperature $\theta_R$ at point $(\vec{x}_R, t_R)$, a fluid temperature $\theta_F$ at time $t_F$ and a temperature $\theta_D$ imposed at the boundary at point $(\vec{y}_D, t_D)$. For clarity, we represent only one fluid cavity.

with

$$
\begin{aligned}
\Theta_S(\vec{x}, t) = \quad & \mathcal{B}_1(p_I^s)\theta_I(\vec{X}_I^s) + (1 - \mathcal{B}_1(p_I^s))\mathcal{B}_2(p_2^s)\theta_R(\vec{X}_R^s, T_R^s) \\
& + (1 - \mathcal{B}_1(p_I^s))(1 - \mathcal{B}_2(p_2^s))\mathcal{B}_3(p_3^s)\theta_F(\vec{Y}_F^s, T_F^s) \\
& + (1 - \mathcal{B}_1(p_I^s))(1 - \mathcal{B}_2(p_2^s))(1 - \mathcal{B}_3(p_3^s))\theta_D(\vec{Y}_D^s, T_D^s)
\end{aligned}
\tag{21}
$$

To simplify the presentation, the definition of the random variables and probabilities involved in this equation are reported to Appendix C.

For given values of $\vec{x}$ and $t$, Algorithm 3 describes the sampling procedure for $\Theta_S$ defined as above and Fig 4 illustrates corresponding typical realizations.

**Algorithm 3**: Sampling algorithm for the random variable $\Theta_S$ defined by Eq (21) assuming the functions $\theta_R$ and $\theta_F$ are known. $\hat{\theta}_S$ is the corresponding realization of the random variable.

```
Sample r₁ uniformly on [0, 1];
if r₁ < pₛᵢ then
  Sample x⃗ᵢ according to the law of X⃗ˢᵢ;
  θ̂ₛ = θᵢ(x⃗ᵢ);
else
  Sample r₂ uniformly on [0, 1];
  if r₂ < pₛ₂ then
    Sample (x⃗ᵣ, τᵣ) according to the law of (X⃗ˢᵣ, Tˢᵣ);
    θ̂ₛ = θᵣ(x⃗ᵣ, τᵣ);
  else
    Sample r₃ uniformly on [0, 1];
    if r₃ < pₛ₃ then
      Sample (y⃗ᶠ, τᶠ) according to the law of (Y⃗ˢᶠ, Tˢᶠ);
      θ̂ₛ = θᶠ(y⃗ᶠ, τᶠ);
    else
      Sample (y⃗ᴅ, τᴅ) according to the law of (Y⃗ˢᴅ, Tˢ);
      θ̂ₛ = θᴅ(y⃗ᴅ, τᴅ);
```

### 4.3 Radiative transfer in fluid and solid sub-domain

We start from the radiative transfer model described by Eq (6d): here phase space is the union of the sets of positions and directions, and boundary conditions are given by the known function $\theta_{R,\partial\Omega_R,\vec{u}}$ on the fictive surface $\partial\Omega_R$. The temperature $\theta$ which appears in the equation is either a fluid temperature, or a temperature in the solid, and results from all the coupling dynamics. As in the two previous paragraphs, we are going to consider first that $\theta$ is known and prescribed so that we can build the probabilization on the uncoupled model.

In agreement with the general form of Eqs (7) and (10), the Green function $g_R \equiv g_R(\vec{x}, \vec{u}|\vec{x'}, \vec{u'})$ associated with the model in Eq (6d) is solution of:

$$\begin{cases} \vec{u}.\vec{\nabla}g_R &+ k_e g_R - k_s \int_{\mathbb{S}^2} p_S(\vec{u}|\vec{u'})\mathrm{d}\vec{u'}g_R = \delta(\vec{x}-\vec{x'})\delta(\vec{u}-\vec{u'}), & \vec{x} \in \overset{\circ}{\Omega}, & \vec{u} \in \mathbb{S}^2 \\[2ex] g_R &= 0\,, & \vec{y} \in \partial\Omega_R, & \vec{u} \in \mathbb{S}^2_+ \end{cases} \quad (22)$$

Radiance temperature in the fluid and solid domains is therefore written as:

$$\begin{aligned} \theta_{R,\vec{u}}(\vec{x},t) &= \int_\Omega \int_{\mathbb{S}^2} g_{R,A}(\vec{x},\vec{u}|\vec{x}_A,\vec{u}_A)\theta(\vec{x}_A,t)\mathrm{d}\vec{u}_A\mathrm{d}\vec{x}_A \\[2ex] &+ \int_{\partial\Omega_R}\int_{\mathbb{S}^2_+} g_{R,\partial\Omega_R}(\vec{x},\vec{u}|\vec{y}_R,\vec{u}_R)\theta_{R,\partial\Omega_R,\vec{u}_R}(\vec{y}_R,t)\mathrm{d}\vec{u}_R\mathrm{d}\vec{y}_R \end{aligned} \quad (23)$$

where $g_{R,A} = g_R(k_e - k_s) = g_R k_a$ stands for the propagator from the temperature in the solid or fluid volumes (Absorption) and $g_{R,\partial\Omega_R} = g_R$ stands for the propagator from the radiative boundary condition on $\partial\Omega_R$.

A random variable whose expectation is the radiance temperature $\theta_{R,\vec{u}}(\vec{x},t)$ is obtained following the proposition stated in the general case (Eqs (7) to (14)):

$$\theta_{R,\vec{u}}(\vec{x},t) = \mathbb{E}[\Theta_{R,\vec{u}}(\vec{x},t)] \quad (24)$$

with

$$\Theta_{R,\vec{u}}(\vec{x},t) = \mathcal{B}(p_A^R(\vec{x},\vec{u}))\theta(\vec{X}_A^R,t) + (1 - \mathcal{B}(p_A^R(\vec{x},\vec{u})))\theta_{R,\partial\Omega_R,\vec{U}_R}(\vec{Y}_R^R,t) \quad (25)$$

To simplify the presentation, the definition of the random variables and probabilities involved in this equation are reported to Appendix D.

A new random variable is defined in order to formulate the temperature $\theta_R(\vec{x},t)$ as an expectation:

$$\Theta_R(\vec{x},t) = \Theta_{R,\vec{U}}(\vec{x},t) \quad (26)$$

where $\vec{U}$ follows a uniform law on the sphere, hence:

$$\theta_R(\vec{x},t) = \int_{\mathbb{S}^2} \frac{1}{4\pi}\mathrm{d}\vec{u}\,\theta_{R,\vec{u}}(\vec{x},t) = \mathbb{E}[\Theta_R(\vec{x},t)] \quad (27)$$

For given values of $\vec{x}$ and $t$, Algorithm 4 describes the sampling procedure for $\Theta_R$ defined as above and Fig 5 illustrates corresponding typical realizations.

### 4.4 Summary

In this section, we have built probabilized forms for heat balance equation inside fluidic cavities, heat balance equation inside solid matrix and radiative transfer equation. This was done

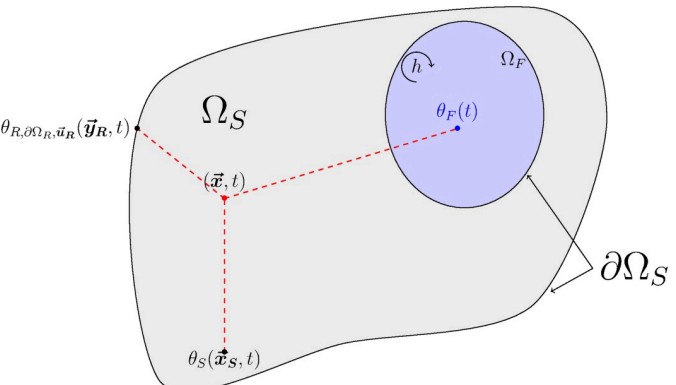

**Fig 5. Representation of three realizations of $\Theta_R$: A radiance temperature imposed on the boundary** $\theta_{R,\partial\Omega_R,\vec{u}_R}$ **at point** $\vec{y}_R$ **(at this point, the boundary** $\partial\Omega_R$ **coincides with** $\partial\Omega_S$**), a solid temperature** $\theta_S$ **at point** $\vec{x}_S$**, and a fluid temperature** $\boldsymbol{\theta_F}$**.** We note $\vec{x}_A \equiv \vec{x}_S$ and $\theta(\vec{x}_A,t) \equiv \theta_S(\vec{x}_S,t)$ if $\vec{x}_A \in \Omega_S$, and $\vec{x}_A \equiv \vec{x}_F$ and $\theta(\vec{x}_A,t) \equiv \theta_F(\vec{x}_F,t) = \theta_F(t)$ if $\vec{x}_A \in \Omega_F$. For clarity, we represent here only one fluid cavity.

by partitioning the model: for each case, an uncoupled form was considered, *i.e* all the variables involved in the coupling between submodels were assumed to be known. Under this assumption, we obtained a random variable whose expectation is the temperature of interest within each submodel, with a corresponding sampling algorithm. Practical implementation of these algorithms is not discussed at this point; it will be the object of Section 7. We focus now on *re-coupling* the three submodels.

**Algorithm 4**: Sampling algorithm for the random variable $\Theta_R$ defined by Eq (27) assuming the function $\theta$ is known (depending on the location, $\theta(\vec{x}_A,t)$ corresponds to a fluid temperature or a solid temperature). $\hat{\theta}_R$ is the corresponding realization of the random variable.

```
Sample 𝒖⃗ according to the law of 𝑼⃗;
Sample r uniformly on [0, 1];
if r < p_A^R(𝒙⃗, 𝒖⃗) then
    Sample 𝒙⃗_A according to the law of 𝑿⃗_A^R;
    θ̂_R = θ(𝒙⃗_A, t);
else
    Sample (𝒚⃗_R, 𝒖⃗_R) according to the law of (𝒀⃗_R^R, 𝑼⃗_R);
    θ̂_R = θ_{R,∂Ω_R,𝒖⃗_R}(𝒚⃗_R, t);
```

## 5 The coupling

Under the decoupling assumption, there exists for each submodel a random variable whose expectation is the temperature we are looking for, with a corresponding sampling algorithm. The key point is that we built formulations in which the re-coupling can now be done solely through sources in Green's sense, *i.e* through inhomogeneous terms in the descriptive equations. It will allow us to solve the whole coupled system by building random walks switching from one algorithm to another; this translates in probabilistic terms how we handle the recursivity of the implicit formulations.

To illustrate this question, the propagator equations are summarized below, keeping only the basic structure of the partitioning/re-coupling (Eq (15) for the fluid temperature, Eq (19)

for the solid temperature, Eqs (27) and (23) for the radiative temperature):

$$
\begin{cases}
\theta_F = g_{F_i,I}\theta_I + \int_{\Delta t \cup \Omega_F} \mathrm{d}\mu g_{F,R}\theta_R + \int_{\Delta t \cup \partial\Omega_F} \mathrm{d}\mu g_{F,S}\theta_S \\[2mm]
\theta_S = \int_{\Omega_S} \mathrm{d}\mu g_{S,I}\theta_I + \int_{\Delta t \cup \Omega_S} \mathrm{d}\mu g_{S,R}\theta_R + \int_{\Delta t \cup \partial\Omega_F^D} \mathrm{d}\mu g_{S,\partial\Omega_F^D}\theta_F + \int_{\Delta t \cup \partial\Omega_D} \mathrm{d}\mu g_{S,\partial\Omega_D}\theta_D \\[2mm]
\theta_R = \int_{\mathbb{S}^2 \cup \Omega \cup \mathbb{S}^2} \mathrm{d}\mu \frac{1}{4\pi} g_{R,A}\theta_{F/S} + \int_{\mathbb{S}^2 \cup \partial\Omega_R \cup \mathbb{S}^2_+} \mathrm{d}\mu \frac{1}{4\pi} g_{R,\partial\Omega_R}\theta_{R,\partial\Omega_R}
\end{cases}
\tag{28}
$$

where $\mu$ is the measure associated to each integration space, $\theta_F$ is the generic form for a fluid temperature whatever the cavity, and $\theta_{F/S}$ stands for the fluid or solid temperature according to the location in the domain.

Each equation in System (28) involves prescribed terms (known initial conditions $\theta_I$, boundary conditions $\theta_D$ or $\theta_{R,\partial\Omega_R}$) and cross-coupling terms between solid, fluid and radiative temperatures, which are space-time functions. Formally this system of integral equations is a Fredholm equation of the second kind for the vector $\vec{\theta} \equiv (\theta_F, \theta_S, \theta_R)$:

$$
\vec{\theta} = \vec{f} + \vec{\mathcal{I}}(\vec{\theta})
\tag{29}
$$

where $\vec{f}$ contains the prescribed terms and $\vec{\mathcal{I}}$ is a linear integral vector operator acting on the temperatures vector $\vec{\theta}$.

By construction of the model in Eq (6) the integral operator $\vec{\mathcal{I}}$ has contracting kernel, which enables to apply the iterated kernel technique to establish solutions in the form of integral Neumann-series. Thanks to this property, solving Eq (29) fits into the standard of the MC method [93–95]. With this approach, contributions of the terms in the infinite Neumann-series expansion are statistically sampled, leading to the algorithmic construction of random walks providing unbiased estimation of the solution. In this regard, solving Fredholm equation of the second kind with MC can be considered as the functional extension of classical MC methods for algebraic linear systems of any dimension [96–98]. From a practical point of view, such methods estimate a probe quantity and in no way the whole field (in continuous cases) or the ensemble of the unknown (in discrete cases). Typically, when solving algebraic linear systems with MC, one of the unknown is estimated without assessing the others, but the implemented random walk statistically *crosses* the entire set of equations in order to ensure the exactness of the result. In the present case, we will estimate the temperature at a given location and time, without having to estimate the entire fields of temperature, thanks to the implementation of random walks switching from one submodel to the other.

## 5.1 Double randomization: A key point

As in MC methods solving Fredholm equations, we base our proposition on the estimation of potentially infinitely nested expectations: for example, the random variable $\Theta_S$, whose expectation is the temperature in the solid, is a function of the temperatures $\theta_{F_i}$ in the fluid cavities, which are themselves expectations of random variables $\Theta_{F_i}$. To understand the recursive mechanics of probabilized coupling through nested expectations, it is interesting to isolate the essential MC property that is classically named *double randomization* [83, 84]. This property is trivial but leads to a subtle gesture that is very powerful because it enables to think the question of the nesting (or the coupling in the present case) locally, *i.e* when the question is raised, at one point of the random walk sampling.

The following elementary illustration contains all the features of the double randomization:

We consider an observable $\theta_1$, written as the expectation of a random variable $B$, which is itself defined as an algebraic linear operator $\mathcal{L}$ on the function $\theta_2$ of a random variable $X$.

$$\begin{cases} \theta_1 = \mathbb{E}(B) \\ B = \mathcal{L}(\theta_2(X)) \end{cases} \tag{30}$$

In this case, the sampling algorithm for $B$ is trivial:

**Algorithm 5**: $b$ is a realization of $B$ in the case where $\theta_2$ is an explicit and known function (Eq (30))

```
Sample x according to the law of X;
b = L(θ₂(x))
```

Double randomization takes place as soon as the function $\theta_2(x)$ itself is expressed as an expectation of another random variable $A(x)$ parametrized by $x$.

$$\theta_2(x) = \mathbb{E}[A(x)] \tag{31}$$

In a naive approach, one could think that it is required to evaluate the expectation of $A(x)$ for each realization $x$ of $X$ in order to be able to use the Algorithm 5 to sample $B$.

Yet, the law of iterated expectations enables to define a new random variable $\tilde{B}$ such that

$$\begin{cases} \theta_1 = \mathbb{E}[\tilde{B}] \\ \tilde{B} = \mathcal{L}(A(X)) \end{cases} \tag{32}$$

which leads to the following algorithm to sample $\tilde{B}$:

**Algorithm 6**: $\tilde{b}$ is a realization of $\tilde{B}$ in the case where $\theta_2$ is defined from an expectation (Eq (31))

```
Sample x according to the law of X;
Sample a according to the law of A(x);
b̃ = L[a]
```

In practice, this double randomization operation is invoked whenever it is necessary to estimate a quantity written in the form of an expectation at a step of random walk sampling. Double randomization is essential in standard MC practice but often, it is not made explicit because the processes which make the algorithmic proposal analogous to physics rely on an intuitive vision that enables to circumvent formalization. For instance, in linear transport physics, when intuitively sampling multiple scattering paths, at each scattering event, the path continues in only one randomly sampled direction: this is the hallmark of double randomization. If, by contrast, probabilization of Fredholm integral equations is the starting point, then double randomization allows for an increased range of flexible application by simply avoiding to address systematically the whole Neumann-series expansion.

This property vanishes as soon as the operators combining the expectations are no longer linear [99–101]. However, recent works [102, 103] have shown that it is possible to extend the proposition to nonlinear cases, by expanding the nonlinear functions as a Taylor-series and then writing each monomial in the series as the product of independent and identically distributed random variables.

## 5.2 A recursive algorithmic approach: Towards a coupled path space

For each equation in System (28) taken independently, a probabilistic version was built by defining random variables whose expectation are the temperatures of interest (Eqs (17), (21)

and (26)). To find the solution of the coupled system with an iterative procedure, we now face the fact that the different functions $\theta$ of the random variables are only known at the boundaries of the overall problem (temporal and spatial); everywhere else they are themselves the expectation of new random variables. Double randomization is here used to address this question. The strength of this approach is that double randomization can be used in a nested manner, as many times as necessary, whenever the situation arises. Hence, random processes are going to interlock recursively until an outcome is found, at a boundary or an initial condition.

The algorithmic translation of this proposition becomes trivial from the algorithms defined for each uncoupled equation (Algorithms 2, 3, 4): the estimation of unknown $\theta$ functions ($\theta_S$, $\theta_R$ or $\theta_F$) at a given location and/or time is simply replaced by a call to the corresponding sampling procedure. Then, the iterative sequence switching from one process to another (based on explicit probabilities) is ended as soon as an initial condition $\theta_I$ or a boundary condition $\theta_D$ or $\theta_{R,\partial\Omega_R}$ is met.

The result of the coupled procedure for a full realization is therefore to generate from the probe at $\vec{x}$ and at a given observation time, a sequence of points that move in space and back in time toward the initial condition. This sequence of points creates a thermal path consisting of a succession of sub-paths associated with each transfer mode. These sub-paths sampled according to Algorithms 2, 3, 4 will be named *convective path*, *conductive path* and *radiative path* respectively. At this stage a sub-path is defined only by its origin and its endpoint.

If $\vec{x}$ is within the solid, the first step in the MC algorithm consists in sampling a conductive path. If $\vec{x}$ is within the fluid, this first step is a convective path. There are also situations where the first step is a radiative path, typically when producing an infrared image by simulating a camera sensor. In all cases, at the end of this sub-path, either the temperature is known and the algorithm stops, or the temperature is unknown and a new path is sampled:

- if the unknown temperature is a solid temperature, the new path is a conductive path within the solid,

- if the unknown temperature is a fluid temperature, the new path is a convective path within the fluid,

- if the unknown temperature is a radiative temperature, the new path is a radiative path that may travel through both the solid and the fluid,

The process is continued until a sub-path ends at a location and time for which the temperature is known. This succession of sampled conductive, convective or radiative paths will be named a *recursive path*. An illustration of such a sequence is proposed in Fig 6.

In the proposition made here, we do not discuss the practicability of sampling the different random variables. In particular cases where propagators (and therefore the probability density functions) have an explicit and known form, building this sampling is undemanding. It leads to an easy and very efficient numerical implementation providing an exact solution in the statistical meaning of the term (this is the case for the model in fluid cavities for example). However, in most situations involving complex geometries, the analytical expressions of the propagators are inaccessible. This situation is often encountered in MC methods and indeed, sampling paths does not always require the functional form of the propagators to be explicit. The following section will focus on this question. The main difficulty will be related to the diffusive component appearing in the energy conservation equation within the solid sub-domain (Eq (6b)). To overcome this difficulty, we get into the theory of stochastic processes, in relation with Feynman-Kac formulation.

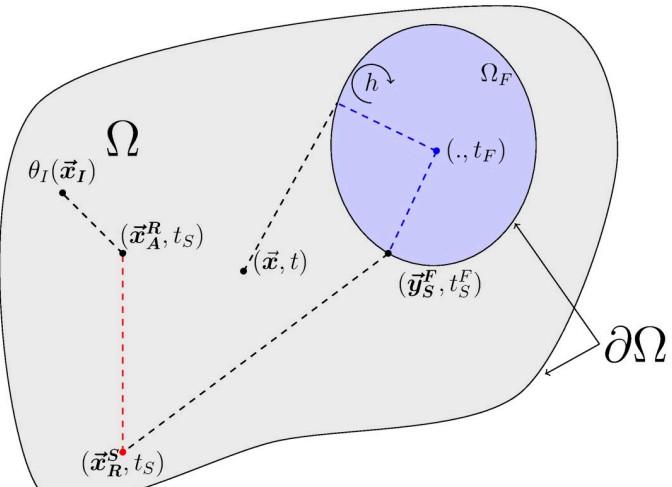

**Fig 6. Illustration of a realization of a recursive path starting from $(\vec{x}, t)$ in the framework of model (6).**
Information first spreads by conduction in the solid domain, until it reaches the fluid domain. Once in the fluid, it
propagates by convection until it reaches the solid boundary at point $(\vec{y}_S^F, t_S^F)$. Back in the solid, information continues
to spread by conduction until reaching a radiative source at $(\vec{x}_R^S, t_S)$. Then it propagates by radiation until being
absorbed in the solid at point $(\vec{x}_A^R, t_S)$, before finally reaching by conduction the point $(\vec{x}_I, t_I)$ where temperature is
known. Through this example recursive path, the contribution to temperature $\theta(\vec{x}, t)$ is the initial condition $\theta_I(\vec{x}_I)$.

## 6 Feynman-Kac approach and the definition of continuous sub-paths

The Green approach of the three previous sections is self-consistent in the definition of a coupled heat transfer MC algorithm. The recursive nature of this algorithm is a way to address the propagator of the overall system from a statistical reading of separate conduction, convection and radiation propagators.

When describing this recursivity, we defined recursive paths made of random successions of conductive paths, convective paths and radiative paths. At this stage, these paths are not yet fully defined, since only the sub-path ending and succession probabilities $p_I^{F_i}, p_R^{F_i}, p_1^S, p_2^S, p_3^S$ and $p_A^R$ have been provided (Eqs (17), (21) and (25)). Hence, the Green formulations of Eqs (15), (19) and (23) define propagators, from the sources (at the boundary and within the domain) to the observation location, but with no time-resolved path-like interpretation.

In this section, the path definition is completed using a stochastic interpretation of the very same physics. Three processes are introduced, $\vec{\mathscr{X}}, \vec{\mathscr{R}}$ and $\vec{\mathscr{U}}$, together with four random variables, $\tau_{\mathscr{X}}, \tau_{\mathscr{R}}, \epsilon_{\mathscr{X}}$ and $\epsilon_{\mathscr{R}}$. They can be used to write a set of three coupled Feynman-Kac formulations of the solid, fluid and radiative temperatures in strict correspondence with Eq (6) and the propagators are built from the statistics of these sub-paths.

$\vec{\mathscr{X}}$ is defined on a domain $\Omega$ that is either a solid or a fluid connex domain, $\Omega \equiv \Omega_S$ or $\Omega \equiv \Omega_F$. The notation $\vec{\mathscr{X}}_p^{\vec{x},t}$ is for $\vec{\mathscr{X}}$ at time $p$, conditioned to reach location $\vec{x}$ at time $t$ (i.e. $\vec{\mathscr{X}}_t^{\vec{x},t} = \vec{x}$). The associated random variable $\tau_{\mathscr{X}}^{\vec{x},t}$ is defined as the time at which $\vec{\mathscr{X}}$ hits the parabolic boundary $(\Omega \times \{t_I\}) \cup (\partial\Omega \times [t_I, t])$.

### 6.1 Conductive path

When located inside a solid domain, $\vec{\mathscr{X}}$ is designed to allow a Feynman-Kac formulation of the solution of Eq (6b) and defines a conductive path. Eq (6b) is a source-diffusion equation

with Robin boundary condition (the particular situation of a Dirichlet condition is, by construction, included in this case), which leads to a Partially Reflected Brownian Motion (PRBM) with $\vec{\mathcal{X}}$ solution of the following Stochastic differential equation (Skorokhod stochastic process, for further details see [104–106]):

$$\mathrm{d}\vec{\mathcal{X}}_p = \mathrm{d}W_p + n(\vec{\mathcal{X}}_p)H_{\partial\Omega_S}(\vec{\mathcal{X}}_p)l_p \tag{33}$$

where $W$ is a three-dimensional Brownian motion, $l_p$ a local boundary time process, $n$ a function on $\Omega_S$ and $H_{\partial\Omega_S}(\vec{x}) = 1$ if $\vec{x} \in \partial\Omega_S$.

## 6.2 Convective path

When located inside a fluid domain, $\vec{\mathcal{X}}$ is designed to allow a Feynman-Kac formulation of the solution of Eq (6c) and defines a convective path. Eq (6c) can be rewritten in probabilistic form (Eqs (15), (17) and Appendix B) which leads to the definition of $\vec{\mathcal{X}}$ as a white noise uniformly distributed on $\Omega_F$ that hits the boundary with a constant rate $\mu = \bar{h}_F \mathscr{S}_F$ ($\bar{h}_F$ is the average $h_F$ over the boundary $\partial\Omega_F$ of area $\mathscr{S}_F$) at locations $\vec{Y}_S^F \equiv \vec{\mathcal{X}}_{\tau_{\mathcal{X}}}$ distributed according to

$$p_{\vec{Y}_S^F}(\vec{y}_S) = \frac{h_F(\vec{y}_S)}{\mu} \tag{34}$$

## 6.3 Radiative path

$\vec{\mathcal{R}}$ is defined on the whole system, i.e. the union of all solid and fluid connex domains, i.e. $\Omega_S \cup \Omega_{F_1} \cup \Omega_{F_2} \cup \Omega_{F_3} \ldots \cup \Omega_{F\infty}$ and $\vec{\mathcal{U}}$ is defined on the unit sphere.

These two processes are designed to allow a Feynman-Kac formulation of the solution of Eq (6d) and define a radiative path. Eq (6d) is a stationary linear Boltzmann equation, which leads to a Stochastic Transport Process, i.e. the standard Markov process of linear transport theory [107, 108]:

$$d\vec{\mathcal{R}}_p = c\vec{\mathcal{U}}_p dp \tag{35}$$

where $c$ is the speed of light and $\vec{\mathcal{U}}$ jumps according to the single scattering phase function at instants given by a Poisson process of rate $\int_0^p k_s\left(\vec{\mathcal{R}}_\nu^{\vec{x},t}\right)d\nu$, that is, the duration between two consecutive collisions is exponentially distributed.

## 6.4 Coupling

These three definitions lead to the following three coupled functional integrals, that are strictly compatible with Eqs (6b), (6c) and (6d):

$\vec{x} \in \Omega_S, t \in [t_I, +\infty[$ :

$$\theta_S(\vec{x}, t) = \mathbb{E}\left[ q_S(\vec{\mathcal{X}}_{\tau_{\mathcal{X}}^{\vec{x},t}}^{\vec{x},t}, \tau_{\mathcal{X}}^{\vec{x},t})\exp\left( -\int_{\tau_{\mathcal{X}}^{\vec{x},t}}^{t} \alpha(\vec{\mathcal{X}}_p^{\vec{x},t}, p)\mathrm{d}p \right) \right.$$
$$\left. + \int_{\tau_{\mathcal{X}}^{\vec{x},t}}^{t} \alpha(\vec{\mathcal{X}}_p^{\vec{x},t}, p)\theta_R(\vec{\mathcal{X}}_p^{\vec{x},t}, p)\exp\left( -\int_p^t \alpha(\vec{\mathcal{X}}_\nu^{\vec{x},t}, \nu)\mathrm{d}\nu \right)\mathrm{d}p \right] \tag{36}$$

$\vec{x} \in \Omega_F, t \in [t_I, +\infty[$ :

$$\theta_F(\vec{x}, t) = \mathbb{E}\left[ q_F(\vec{\mathcal{X}}^{\vec{x},t}_{\tau^{\vec{x},t}_{\mathcal{X}}}, \tau^{\vec{x},t}_{\mathcal{X}}) \exp\left( - \int_{\tau^{\vec{x},t}_{\mathcal{X}}}^t \alpha(\vec{\mathcal{X}}^{\vec{x},t}_p, p) dp \right) \right.$$

$$\left. + \int_{\tau^{\vec{x},t}_{\mathcal{X}}}^t \alpha(\vec{\mathcal{X}}^{\vec{x},t}_p, p) \theta_R(\vec{\mathcal{X}}^{\vec{x},t}_p, p) \exp\left( - \int_p^t \alpha(\vec{\mathcal{X}}^{\vec{x},t}_v, v) dv \right) dp \right] \quad (37)$$

$t \in [t_I, +\infty[$ :

$$\theta_R(\vec{x}, t) = \mathbb{E}\left[ \tilde{\theta}_R(\vec{\mathcal{R}}^{\vec{x},t}_{\tau^{\vec{x},t}_{\mathcal{R}}}, \vec{\mathcal{U}}^{\vec{x},t}_{\tau^{\vec{x},t}_{\mathcal{R}}}, \tau^{\vec{x},t}_{\mathcal{R}}) \exp\left( - \int_{\tau^{\vec{x},t}_{\mathcal{R}}}^t \beta(\vec{\mathcal{R}}^{\vec{x},t}_p, p) dp \right) \right.$$

$$\left. + \int_{\tau^{\vec{x},t}_{\mathcal{R}}}^t \beta(\vec{\mathcal{R}}^{\vec{x},t}_p, p) \theta_{F/S}(\vec{\mathcal{R}}^{\vec{x},t}_p, p) \exp\left( - \int_p^t \beta(\vec{\mathcal{R}}^{\vec{x},t}_v, v) dv \right) dp \right] \quad (38)$$

with

$$q_S(\vec{x}, t) \equiv \begin{cases} \theta_I(\vec{x}) & \text{if } t = t_I, \ \vec{x} \in \Omega_S \\ \theta_F(\vec{x}, t) & \text{if } t > t_I, \ \vec{x} \in \partial\Omega_F^D \\ \theta_D(\vec{x}, t) & \text{if } t > t_I, \ \vec{x} \in \partial\Omega_D \end{cases} \quad (39)$$

$$q_F(\vec{x}, t) \equiv \begin{cases} \theta_I(\vec{x}) & \text{if } t = t_I, \ \vec{x} \in \Omega_F \\ \theta_S(\vec{x}, t) & \text{if } t > t_I, \ \vec{x} \in \partial\Omega_F \end{cases} \quad (40)$$

$$\alpha = \frac{\zeta}{\rho C} \quad (41)$$

$$\beta = k_a c \quad (42)$$

where $\tilde{\theta}_R \equiv \theta_{R,\partial\Omega_R,\vec{u}}$ is the directional radiative temperature in incoming directions at the limit of the composite domain, and $\theta_{F/S}$ is either the temperature of the fluid $\theta_F$ or the temperature of the solid $\theta_S$, depending on the position.

These sub-path statistics then achieve the objective of completing the spatio-temporal description of section 4 on decoupled models for each of the three heat transfer modes separately.

## 6.5 Switching from one mode to the other

Of course, Eqs (36), (37) and (38) are coupled: $\theta_R$ appears as an integrated source in Eqs (36) and (37) and $\theta$ as an integrated source in Eq (38). To recover the recursive path description of Section 5, this coupling must be translated into thermal paths switching from one mode to the other (whether in the domain or at an interface). Following a very standard MC approach, this is achieved by defining random variables $\epsilon_{\mathcal{X}}$ and $\epsilon_{\mathcal{R}}$ that turn the source integrations into expectations. When conditionned by $\vec{x}$ and $t$, $\epsilon^{\vec{x},t}_{\mathcal{X}}$ is defined on $[\tau^{\vec{x},t}_{\mathcal{X}}, t]$ with a probability

density

$$p_{\epsilon_{\mathscr{X}}^{\vec{x},t}}(\epsilon) = \exp\left( -\int_{\tau_{\mathscr{X}}^{\vec{x},t}}^{t} \alpha(\vec{\mathscr{X}}_{p}^{\vec{x},t}, p) \mathrm{d}p \right) \delta(\epsilon - \tau_{\mathscr{X}}^{\vec{x},t}) + \alpha(\vec{\mathscr{X}}_{\epsilon}^{\vec{x},t}, \epsilon) \exp\left( -\int_{\epsilon}^{t} \alpha(\vec{\mathscr{X}}_{p}^{\vec{x},t}, p) \mathrm{d}p \right) \quad (43)$$

Similarly, $\epsilon_{\mathscr{R}}^{\vec{x},t}$ is defined on $[\tau_{\mathscr{R}}^{\vec{x},t}, t]$ with

$$p_{\epsilon_{\mathscr{R}}^{\vec{x},t}}(\epsilon) = \exp\left( -\int_{\tau_{\mathscr{R}}^{\vec{x},t}}^{t} \beta(\vec{\mathscr{R}}_{p}^{\vec{x},t}, p) \mathrm{d}p \right) \delta(\epsilon - \tau_{\mathscr{R}}^{\vec{x},t}) + \beta(\vec{\mathscr{R}}_{\epsilon}^{\vec{x},t}, \epsilon) \exp\left( -\int_{\epsilon}^{t} \beta(\vec{\mathscr{R}}_{p}^{\vec{x},t}, p) \mathrm{d}p \right) \quad (44)$$

Reporting these definitions into Eqs (36), (37) and (38) leads to

$$\vec{x} \in \Omega_{S}, t \in [t_{I}, +\infty[ \ : \theta_{S}(\vec{x}, t) = \ \mathbb{E}\left[ \ r_{S}(\vec{\mathscr{X}}_{\epsilon_{\mathscr{X}}^{\vec{x},t}}^{\vec{x},t}, \epsilon_{\mathscr{X}}^{\vec{x},t}) \ \right] \quad (45)$$

$$\vec{x} \in \Omega_{F}, t \in [t_{I}, +\infty[ \ : \theta_{F}(\vec{x}, t) = \ \mathbb{E}\left[ \ r_{F}(\vec{\mathscr{X}}_{\epsilon_{\mathscr{X}}^{\vec{x},t}}^{\vec{x},t}, \epsilon_{\mathscr{X}}^{\vec{x},t}) \ \right] \quad (46)$$

$$t \in [t_{I}, +\infty[ \ : \theta_{R}(\vec{x}, t) = \ \mathbb{E}\left[ \ r_{R}(\vec{\mathscr{R}}_{\epsilon_{\mathscr{R}}^{\vec{x},t}}^{\vec{x},t}, \vec{\Omega}_{\epsilon_{\mathscr{R}}^{\vec{x},t}}^{\vec{x},t}, t) \ \right] \quad (47)$$

with

$$r_{S}(\vec{x}, t) \ \equiv \ \begin{cases} \theta_{I}(\vec{x}) & \text{if } t = t_{I}, \ \vec{x} \in \Omega_{S} \\ \theta_{F}(\vec{x}, t) & \text{if } t > t_{I}, \ \vec{x} \in \partial\Omega_{F}^{D} \\ \theta_{D}(\vec{x}, t) & \text{if } t > t_{I}, \ \vec{x} \in \partial\Omega_{D} \\ \theta_{R}(\vec{x}, t) & \text{if } t > t_{I}, \ \vec{x} \in \Omega_{S} \end{cases} \quad (48)$$

$$r_{F}(\vec{x}, t) \ \equiv \ \begin{cases} \theta_{I}(\vec{x}) & \text{if } t = t_{I}, \ \vec{x} \in \Omega_{F} \\ \theta_{S}(\vec{x}, t) & \text{if } t > t_{I}, \ \vec{x} \in \partial\Omega_{F} \\ \theta_{R}(\vec{x}, t) & \text{if } t > t_{I}, \ \vec{x} \in \Omega_{F} \end{cases} \quad (49)$$

$$r_{R}(\vec{x}, \vec{\omega}, t) \ \equiv \ \begin{cases} \tilde{\theta}_{R}(\vec{x}, \vec{\omega}, t) & \text{if } \vec{x} \in \partial\Omega_{R} \\ \theta_{S}(\vec{x}, t) & \text{if } \vec{x} \in \Omega_{S} \\ \theta_{F}(\vec{x}, t) & \text{if } \vec{x} \in \Omega_{F} \end{cases} \quad (50)$$

Eqs (45) to (50) allow to find the same recursive structure as the one described in Section 5:

- In Eq (45) the solid temperature is defined as an expectation that involves the fluid temperature at the boundary, via $\theta_{F}$ in Eq (48).

- In Eq (46) the fluid temperature is defined as an expectation that involves the solid temperature at the boundary, via $\theta_{S}$ in Eq (49).

- Both equations involve the radiative temperature of Eq (47), itself an expectation that involves both the solid and fluid temperatures, via $\theta_{S}$ or $\theta_{F}$ in Eq (50).

Now, the recursivity is expressed in terms of processes, such that the physical picture of randomly alternating conductive, convective and radiative paths is justified.

Through coupled stochastic processes, we have then developed a probabilized path space that makes the propagation viewpoint of the previous section operational. Scanning these paths according to the laws of the corresponding stochastic processes leads to strict sampling of the random variables defined by the Eqs (17), (21) and (25). Fig 7 illustrates both visions for the realization of a path. A MC algorithm of the coupled problem can then be designed based on the recursive sampling of the sub-paths defined by stochastic processes. Algorithm 7 gives the corresponding algorithmic prototype.

## 6.6 Sampling paths

In Algorithm 7, the "*Sample a convective path starting at* $(\vec{x}, t)$" part does not raise any specific issue, and has been described in the Algorithm 2.

We can simply mention that the $\Theta_{F_I}$ random variable can be sampled using the null-collision technique [109–111] as soon as the convective exchange coefficient $h_F$ is spatially heterogeneous. The "*Sample a radiative path $\gamma$ starting at location $\vec{x}_R$ at time $t_R$*" part consists in the realization of a stationary radiative path, as described in the previous section by "the standard Markov process of linear transport theory". The realization of such a path in the presence of an absorbing, emitting and scattering medium has been widely described in the literature (see for instance [3, 4, 112]). Numerous radiative transfer simulation codes have implemented this path sampling technique, and various nuances and subtleties are presented in reference books [113–116].

As mentioned above, the main remaining issue for an efficient algorithmic implementation in a confined environment is to generate Brownian trajectories coupled to a radiative source field in a solid medium. We present our choice of implementation in the next section.

**Algorithm 7**: The complete recursive algorithm evaluating temperature at location $\vec{x}^*$ and time $t^*$ with a full conduction/convection/radiation coupling. $\vec{x}^*$ and $t^*$ may be within the solid or within the fluid. $N$ recursive paths are sampled, starting at $\vec{x}^*$, backward in time from $t^*$. The estimator is $m$ and $s$ is its statistical uncertainty. The points where coupling operates are stressed by the keyword *coupling*.

```
sum = 0;
sumOfSquares = 0;
foreach recursive path i in 1:N do
  Set x = x* and t = t*;
  Set recursion to true;
  while recursion do
    case (x is within the solid) do
      Sample a conductive path starting at (x,t) (see 6.1 and 7);
    case (x is within the fluid) do
      Sample a convective path starting at (x,t) (see 6.2 and algo 2);
    case (the path ends at an intial condition) do
      Get the location xI of the end of the path;
      w = θI(xI);
      Set recursion to false;
    case (the path ends at a radiative source) do
      Get the location xR and time tR of the end of the path (coupling);
      Sample a radiative path γ starting at location xR at time tR (see
6.3);
      Get the location xγ and the direction ωγ of the end of the radia-
tive path;
      if (xγ is at a radiative limit) then
```

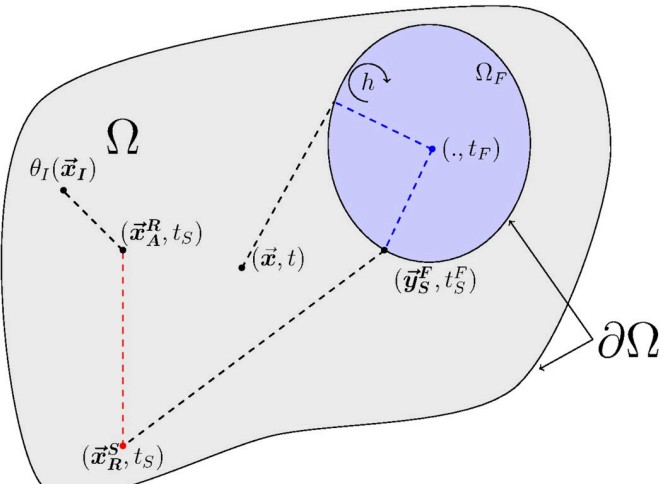

(a) Green functions method

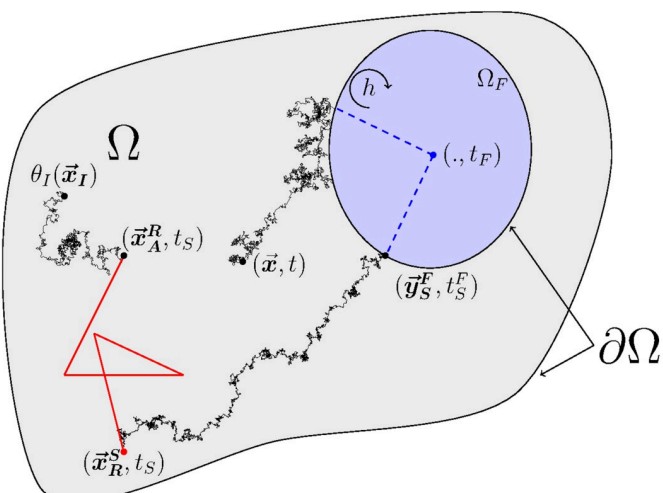

(b) Stochastic process method

**Fig 7.** (a) matches Fig 6 for which only the points corresponding to the end of sub-paths (coupled propagators) are defined. (b) illustrates the whole path for each transfer mode, starting from the observation point $(\vec{x}, t)$ until finding a prescribed temperature (here a temperature at the initial condition at $\vec{x}_I$). Brownian paths are black lines, radiative paths red lines and convective paths by blue dotted lines. The illustrated sequence is: conduction → convection → conduction → radiation → conduction. While only the beginning and the end of each sub-path is shown in (a), a detailed example of the sub-path for each transfer mode is displayed in (b). The standard illustration (see [117]) of the multi-scattering radiative path results from the iterative sampling of scattering free path lengths and propagation directions. (a) Green functions method and (b) Stochastic process method.

```
            w = θ̃(x⃗_γ, ω⃗_γ, t_R);
            Set recursion to false;
        else
            Set x⃗ = x⃗_γ and t = t_R (coupling);
    case (the path ends at a boundary) do
        Get the location x⃗_C and time t_C of the end of the path;
        if (the solid or fluid temperature θ is known at the end of the
path)
            then
                w = θ(x⃗_C, t_C);
                Set recursion to false;
            else
                (see 5.2)
                case (the path is a conductive path) do
                    Set x⃗_C as belonging to the fluid;
                case (the path is a convective path) do
                    Set x⃗_C as belonging to the solid;
                Set x⃗ = x⃗_C and t = t_C (coupling);
    sum = sum + w;
    sumOfSquares = sumOfSquares + w²;
m = sum/N;
s = (1/√N)(sumOfSquares/N − m²);
```

## 7 The practice of sampling Brownian motion with confinement and radiation coupling

The issue of sampling Brownian motion in confined environments with heterogeneous sources is well documented in the literature and is the subject of active research (the Appendix E gives an overview of the most popular approaches to confined Brownian motion). Here we make an alternative choice which is not directly derived from the most standard first passage approaches. It is motivated by the desire to stay as close as possible to path-sampling procedures that are compatible with the efficient ray-tracing techniques developed by the computer graphics community. Our proposal, denoted *δ-sphere random walk for conductive paths*, can be summarized as follows:

1. The diffusion equation is transformed by approximating the Laplacian term by its finite difference version while remaining entirely continuous.

2. Near the boundaries the random walk is adjusted to guarantee a certain level of accuracy, the scheme being exact for linear temperature profiles at steady state.

3. The continuity of the heat flux that ensures the coupling condition at the interfaces is treated with the same level of approximation as the steps described above.

The advantage of reformulating the model with this set of approximations is that once the probabilization is done, it can be solved exactly in the strict MC sense. It allows to separate the approximations of very different nature: on the one hand, the part associated with the rewriting of the model and, on the other hand, the statistical uncertainty of the unbiased estimator of the corresponding expectation. The result in terms of ray tracing and trajectories is illustrated in Fig 8. Let us note in particular two rays of opposite directions at each step of the conductive random walk, jumps of variable size near the boundaries and standard multiple scattering trajectories for the radiative part. We develop in this paragraph the theoretical considerations that lead to this particular scheme. In order to justify the complete random walk scheme in a didactic way, we will separate the steps as follows:

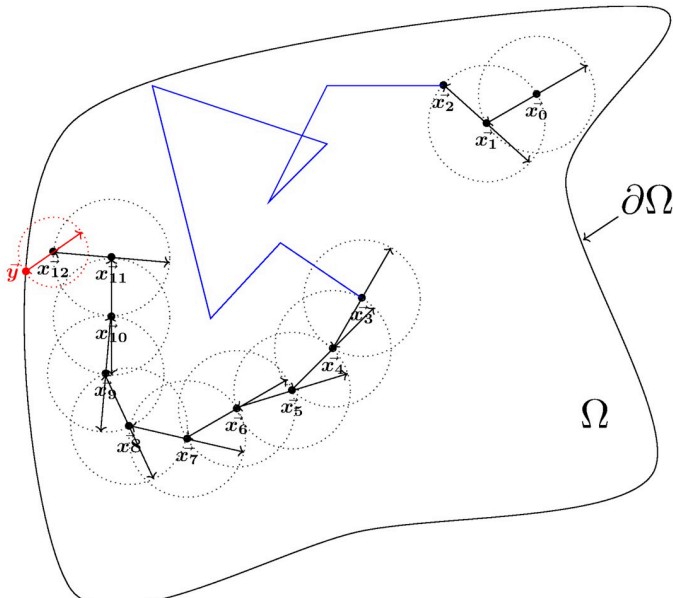

**Fig 8. Illustration of thermal path sampling in a confined environment with a $\delta$-sphere random walk for conductive paths.** The bidirectional arrows represent the fact that an intersection test must be performed in two opposite directions at each jump, in order to obtain the local $\delta$ step value. In red: the walking step is smaller than in the rest of the field, and the walk ends exactly at the boundary. At position $\vec{x}_2$ the conductive path switches to a radiative path until position $\vec{x}_3$.

- Approximations of conducto-radiative coupled system in an infinite medium.

- Modifications to the scheme to account for confinement (essentially with consequences near the boundaries).

- Additional steps related to boundary coupling conditions

## 7.1 The approximation of the coupled conducto-radiative system in an infinite medium

We start with the model 6b rewritten in an infinite geometric space of dimension $n$, with constant thermophysical properties:

$$\begin{cases} \rho C \partial_t \theta_S = \lambda \Delta \theta_S + \zeta(\theta_R - \theta_S) & , (t, \vec{x}) \in ]t_I, +\infty[\times \mathbb{R}^n \\ \theta_S(\vec{x}, t_I) = \theta_I(\vec{x}, t_I) & , \vec{x} \in \mathbb{R}^n \end{cases} \tag{51}$$

From previous sections, $\theta_R$ (required for the radiative coupling) can be expressed from a process that completely defines a path space over which it is possible to formally write the temperature as an expectation:

$$\theta_R(\vec{x}, t) = \int_{\mathscr{D}_\Gamma} p_\Gamma(\gamma) \mathrm{d}\gamma \, \theta_S(\vec{x}_\gamma, t) \quad , (t, \vec{x}) \in ]t_I, +\infty[\times \mathbb{R}^n \tag{52}$$

in which $\mathscr{D}_\Gamma \equiv \mathscr{D}_\Gamma(\vec{x})$ represents the radiative path space of origin $\vec{x}$ over which the path random variable $\Gamma \equiv \Gamma(\vec{x})$ of density $p_\Gamma$ is defined. It is worth noting that the temperature retained at the end of the $\gamma$ path, $\vec{x}_\gamma$, is taken with no modification of date $t$, which means that radiative processes are considered at a stationary state, because they are very fast compared to

the other processes involved. Sampling this path space is straightforward using MC methods: sampling of the $\Gamma$ random variable is easy no matter the complexity of the underlying radiative physics (for instance, taking into account multiple scattering or multiple reflections in all kinds of situations).

The only required approximation in Eq (51) consists in replacing the Laplacian operator by its centered finite differences counterpart. In order to obtain an expression that does not depend on a given cartesian basis, the finite difference is averaged over all possible basis orientations. The retained approximate model is finally:

$$
\begin{cases}
\rho C \partial_t \tilde{\theta}_S = \lambda \left( \dfrac{2n \int_{\mathbb{S}^{n-1}} \dfrac{1}{\mathscr{S}_{n-1}} \mathrm{d}\vec{u}\,\tilde{\theta}_{S,\vec{u}} - 2n\tilde{\theta}_S}{\delta^2} \right) + \zeta(\tilde{\theta}_R - \tilde{\theta}_S) & , (t,\vec{x}) \in ]t_I, +\infty[\times\mathbb{R}^n \\[2em]
\tilde{\theta}_S(\vec{x}, t_I) = \theta_I(\vec{x}) & , \vec{x} \in \mathbb{R}^n \\[1em]
\tilde{\theta}_R(\vec{x}, t) = \displaystyle\int_{\mathscr{D}_\Gamma} p_\Gamma(\gamma)\mathrm{d}\gamma\,\tilde{\theta}_S(\vec{x}_\gamma, t) & , (t,\vec{x}) \in ]t_I, +\infty[\times\mathbb{R}^n
\end{cases}
\tag{53}
$$

where $\mathscr{S}_{n-1}$ is the surface of sphere $\mathbb{S}^{n-1}$ in dimension $n$ and $\tilde{\theta}_{S,\vec{u}} \equiv \tilde{\theta}_{S,\vec{u}}(\vec{x}, t) = \tilde{\theta}_S(\vec{x} + \delta\vec{u}, t)$. It was chosen to use the $\tilde{\theta}_S$ notation for the temperature that is a solution of Eq (53) to specifically mention its dependence to the $\delta$ parameter.

In Appendix F, the model 53 is analytically studied in an infinite medium. Its consistency with the exact model of Eqs (51) and (52) is discussed, as well as the convergence of the solution $\tilde{\theta}_S$ towards $\theta_S$.

## 7.2 Walk on $\delta$-sphere without radiative coupling

In order to understand the approximation proposition, this paragraph first describes the random walk in pure diffusion (i.e. on the only mechanism that is approximated in the field, the radiation being treated in an exact manner). The first equation of (53) without the radiative term (which is the case when $\zeta = 0$) can easily be reformulated as a second kind Fredholm integral:

$$
\tilde{\theta}_S(\vec{x}, t) = \int_0^{+\infty} p_T(\tau)\mathrm{d}\tau \left\{ \begin{array}{l} \mathcal{H}(\tau - t)\theta_I(\vec{x}) \\[1em] + \quad \mathcal{H}(t - \tau)\displaystyle\int_{\mathbb{S}^{n-1}} p\vec{U}(\vec{u})\mathrm{d}\vec{u}\,\tilde{\theta}_S(\vec{x} + \delta\vec{u}, t - \tau) \end{array} \right\}
\tag{54}
$$

That can be re-written as an expectation:

$$
\tilde{\theta}_S(\vec{x}, t) = \mathbb{E}[\mathcal{H}(T - t)\theta_I(\vec{x}) + \mathcal{H}(t - T)\mathbb{E}[\tilde{\theta}_S(\vec{x} + \delta\vec{U}, t - T)]]
\tag{55}
$$

where $p_T$ is the density probability of $T$ that follows an exponential law of parameter $2nD/\delta^2$, where $D = \frac{\lambda}{\rho C}$ is the thermal diffusivity of the material and $p\vec{U}$ is the density probability of the $\vec{U}$ random variable (uniform over the sphere of dimension $n - 1$). Notation $\mathcal{H}(x)$ stands for the Heaviside function, that takes a value of 1 for $x > 0$ and a value of 0 for $x < 0$.

Temperature $\tilde{\theta}_S$ at position $\vec{x}$ and at time $t$ is expressed as the expected value of a linear function of an expectation; the double randomization principle can then be invoked in order

to express the temperature as a unique expectation.

$$\tilde{\theta}_S(\vec{x}, t) \quad = \mathbb{E}[\theta_I(\vec{X}_N)] \tag{56}$$

with:

- $\vec{X}_N = \vec{x} + \delta \sum_{i=0}^{N} \vec{U}_i$, $\vec{U}_0 \equiv \vec{0}$, $T_0 = 0$,

- $N = \min\left\{n \in \mathbb{N}; \sum_{i=0}^{n} T_i > t - t_I\right\}$

- $(T_i)_{i \in \{0,\ldots,N\}}$ and $(\vec{U}_i)_{i \in \{0,\ldots,N\}}$ two series of independent and identically distributed random variables (IID) respectively for $T$ and $\vec{U}$.

It should be noted that $\vec{X}_N$ can be read as a process that replaces the Brownian process $W$ that was defined in the previous section. Eq (56) is a Feynman-Kac equation in the study case. Algorithm 8 gives the simple way in which the sampling of $\theta_I(\vec{X}_N)$ is done for given $\vec{x}$ and $t$.

**Algorithm 8**: $\delta$-sphere algorithm in infinite medium without coupling. $w$ is the generic notation for a realization of the sampled random variable

```
while t > t_I do
    Sample τ according to the law of T;
    t = t - τ;
    if t < t_I then
        w = θ_I(x⃗);
    else
        Sample u⃗ according to the law of U⃗;
        x⃗ = x⃗ + δu⃗;
```

## 7.3 Walk on $\delta$-sphere with radiative coupling

Eq (53) is now considered with its radiative coupling term. From a formal point of view, there is no additional difficulty to obtain a version of this equation as a second kind Fredholm equation:

$$\tilde{\theta}_S(\vec{x}, t) = \int_0^{+\infty} p_T(\tau) d\tau \left\{ \begin{array}{l} \mathcal{H}(\tau > t - t_I)\theta_I(\vec{x}) \\ + \quad \mathcal{H}(\tau < t - t_I)\left( p_C \int_{\mathbb{S}^{n-1}} p\vec{U}(\vec{u}) d\vec{u} \tilde{\theta}_S(\vec{x} + \delta\vec{u}, t - \tau) + p_R \int_{\mathscr{D}_\Gamma} p_\Gamma(\gamma) d\gamma \tilde{\theta}_S(\vec{x}_\gamma, t - \tau) \right) \end{array} \right\} \tag{57}$$

That can be re-written as an expectation:

$$\tilde{\theta}_S(\vec{x}, t) = \mathbb{E}\left[ \begin{array}{l} \mathcal{H}(T - t)\theta_I(\vec{x}) \\ + \quad \mathcal{H}(t - T)(\mathcal{B}(p_C)\mathbb{E}[\tilde{\theta}_S(\vec{x} + \delta\vec{U}, t - T)] + (1 - \mathcal{B}(p_C))\mathbb{E}[\tilde{\theta}_S(\vec{x}_\Gamma, t - T)]) \end{array} \right] \tag{58}$$

where $p_T$ is the density probability of $T$ that follows an exponential law of parameter $(2n\lambda + \zeta\delta^2)/(\rho C\delta^2)$; $p_C = (2n\lambda)/(2n\lambda + \zeta\delta^2)$, $p_R = 1 - p_C$ and $p\vec{U}$ is the probability density of the $\vec{U}$ random variable (uniform over the sphere of dimension $n - 1$). $\mathcal{B}(p_C)$ is a Bernouilli random variable of parameter $p_C$.

Similarly to the no-radiation case, the double randomization principle is used in order to evaluate this expectation by MC. It consists in writing:

$$\tilde{\theta}_S(\vec{x}, t) \quad = \mathbb{E}[\theta_I(\vec{Y}_N)] \tag{59}$$

The random variable $\vec{Y}_N$ is not formally defined here; the Algorithm 9 for the sampling of $\theta_I(\vec{Y}_N)$ for given $\vec{x}$ and $t$ is sufficient to clarify its meaning.

**Algorithm 9**: $\delta$-sphere algorithm in infinite medium with radiative coupling. $w$ is the generic notation for a realization of the sampled random variable.

```
while t > t_I do
    Sample τ according to the law of T;
    t = t - τ;
    if t < t_I then
        w = θ_I(x⃗);
    else
        Sample r uniformly on [0, 1];
        if r < p_C then
            Sample u⃗ according to the law of U⃗;
            x⃗ = x⃗ + δu⃗;
        else
            Sample γ according to the law of Γ(x⃗);
            x⃗ = x⃗_γ;
```

**Property of our $\delta$-sphere scheme**. At steady state without radiative coupling, Eq 54 becomes $\tilde{\theta}_S(\vec{x}, t) = \int_{\mathbb{S}^{n-1}} p\vec{U}(\vec{u}) \mathrm{d}\vec{u} \, \tilde{\theta}_S(\vec{x} + \delta\vec{u})$ which is exact since $\theta_S$ is an harmonic function solution of $\Delta\theta_S = 0$ (take $\zeta = 0$ and $\partial_t\theta_S = 0$ in Eq 51). The error committed in general cases, as a function of the $\delta$ value, is provided in Appendix F.

## 7.4 The approximation of conductive path near the boundary

The $\delta$-sphere random walk has been detailed in an infinite medium. Coupling with radiative transfer does not require any approximation, nor does it introduce any additional difficulty. Remains the question of approximating the Brownian walk in a confined medium, more precisely, how the diffusive random walks deal with boundaries. As in any "Walk on Sphere" method [28, 31, 118, 119], the $\delta$-sphere walk never really ends at a boundary. With a small enough value of $\delta$, it could be considered that, when the random walk crosses the boundary, the intersection position is the final position of the path. An alternative to this trivial solution is proposed here, that reduces the numerical error for a given value of the walking step.

We keep the constraint of using only random walks built upon ray-surface intersections, since our $\delta$-sphere random walk aims at using computer sciences methods for identifying the intersection between a ray and a scene defined by a huge number of geometrical primitives.

To deal with the boundary, we propose a scheme that is exact in the case of linear temperature profiles at stationnary state.

To that purpose, the value of the walking step is adjusted along direction $\vec{u}$:

$$\delta \equiv \delta(\vec{x}, \vec{u}) = \min\{\delta_{\mathrm{ref}}, \delta_{\partial\Omega_S}(\vec{x}, \vec{u})\}$$

where $\delta_{\mathrm{ref}}$ represents the maximum step of the random walk, and $\delta_{\partial\Omega_S}(\vec{x}, \vec{u})$ is the distance to the closest boundary in directions $\vec{u}$ or $-\vec{u}$. Indeed, any linear function $\theta(\vec{x}) = \vec{a}.\vec{x} + b$ satisfies

$$\theta(\vec{x}) = \frac{\theta(\vec{x} + \delta\vec{u}) + \theta(\vec{x} - \delta\vec{u})}{2} \tag{60}$$

In the case when the temperature profile is not globally linear, a value of the $\delta$ parameter that locally ensures a linear profile can be found in almost all cases; this makes the $\delta$-sphere random walk a good approximation, including during non-stationary phases.

From an algorithmic point of view, the proposition is rather straightforward. After sampling a direction $\vec{u}$ and before enacting the displacement, the value of the random walk step has to be evaluated by testing the distance to the boundary in both directions $\vec{u}$ and $-\vec{u}$. The

random walk will therefore always use a value $\delta_{\text{ref}}$ for positions far from the boundary; and for positions that are close to the boundary, the value of the walking step will be automatically reduced (see Fig 8). In the spatial sub-domains where $\delta$ is lower than $\delta_{\text{ref}}$, the random walk will therefore statistically stop at the boundary half the time. If the temperature of the boundary is known (Dirichlet limit condition), the random walk ends at the boundary. Otherwise (Robin limit condition), a specific treatment must be performed, that is described in the following paragraph.

It should be emphasized that Eq (55) is still perfectly valid, even in the presence of boundaries. The main difference in this case is that, in a confined medium, the $\delta$ parameter and the random variable $T$ both depend of the random variable $\vec{U}$. This will translate in terms of algorithm by the fact that the first sampling must be performed for random variable $\vec{U}$. The algorithm for sampling random variables for $\vec{x}$ and $t$ is described in Algorithm 10.

**Algorithm 10**: $\delta$-sphere algorithm in bounded domain without radiative coupling. $w$ is the generic notation for a realization of the sampled random variable.

```
while t > tᵢ or x⃗∉∂Ωₛ do
    Sample u⃗ accordind to the law of U⃗;
    Compute the distances d± to the boundary in the ±u⃗ directions;
    δ = min(δ_ref, d±);
    Sample τ according to the law of T ~ ℰ(2nD/δ²);
    t = t - τ;
    if t < tᵢ then
        w = θᵢ(x⃗);
    else
        x⃗ = x⃗ + δu⃗;
        if x⃗ ∈ ∂Ωₛ then
            w = θ_∂Ωₛ(x⃗,t);
```

## 7.5 Interface conditions and flux continuity

The temperature at the boundary is generally unknown, except in the particular case of a Dirichlet condition (set temperature). When the temperature is unknown, a discretized version of the flux continuity relation for the interface between two media is used, in order to express the boundary temperature as an expectation. The double randomization technique then makes it possible to continue generating the recursive thermal path, as previously shown.

The continuity of the surface flux density over an interface with a Robin condition is written as in Eq (6b):

$$\lambda \vec{n}.\vec{\nabla}\theta_S = -\lambda \frac{\partial \theta_S}{\partial n} = h_F(\theta_F - \theta_S) \qquad , (t,\vec{x}) \in ]t_I, +\infty[ \times \partial\Omega_S^D \tag{61}$$

where $\vec{n}$ is the incoming normal at the surface of the solid and $\theta_F$ is the temperature of the fluid.

As for the field approximation, the normal derivative at the boundary is translated into its finite difference counterpart:

$$\lambda \frac{\tilde{\theta}_S(\vec{x},t) - \tilde{\theta}_S(\vec{x} + \delta_b\vec{N}, t)}{\delta_b} = h_F\left(\theta_F(t) - \tilde{\theta}_S(\vec{x},t)\right) \qquad , (t,\vec{x}) \in ]t_I, +\infty[ \times \partial\Omega_S^D \tag{62}$$

From which the boundary temperature is obtained:

$$\tilde{\theta}_S(\vec{x},t) = p_{\delta_b}\tilde{\theta}_S(\vec{x} + \delta_b\vec{n}, t) + p_F\theta_F(t) \qquad , (t,\vec{x}) \in ]t_I, +\infty[ \times \partial\Omega_S^D \tag{63}$$

with $p_{\delta_b} = \frac{\frac{\lambda}{\delta_b}}{\frac{\lambda}{\delta_b}+h_F}$ and $p_F = \frac{h_F}{\frac{\lambda}{\delta_b}+h_F}$.

$\delta_b$ is a numerical parameter just like $\delta$. In some cases it may be useful to be able to impose $\delta_b$ and $\delta$ separately but in many applications $\delta_b = \delta$ is a relevant choice.

Expression 63 is interpreted as the expectation of a random variable $\tilde{\Theta}_{int}(\vec{x}, t)$:

$$\tilde{\theta}_S(\vec{x}, t) = \mathbb{E}[\tilde{\Theta}_{int}(\vec{x}, t)]$$

with:

$$\tilde{\Theta}_{int}(\vec{x}, t) = \mathcal{B}(p_{\delta_b})\tilde{\theta}_S(\vec{x} + \delta_b\vec{n}, t) + (1 - \mathcal{B}(p_{\delta_b}))\theta_F(t)$$

It was previously shown that the temperature at any position in the solid and the temperature of the fluid can be interpreted as the expectation of well defined random variables. The double randomization technique is then used once again in order to deal with nested expectations; by doing so, the thermal path continues either in the fluid or in the solid, at a distance $\delta_b$ from the boundary.

## 7.6 The path space with random walk on $\delta$-sphere

All items are now available in order to completely define the thermal path space. The Brownian process in the field and the connecting condition between the solid and fluid are the only mechanisms that are approximated. Other processes, as well as all couplings in the field, are treated in an exact manner.

Fig 9 completes Fig 7 by adding a representation of the conductive paths under the approximation of $\delta$-sphere random walk.

## 8 Conclusions and outlooks

The proposal of the present text combines several points of view over MC methods with one leading intention: benefiting from the solid foundations laid in a vast literature. As far as propagation, stochastic processes and integral relations are concerned, the formal background is well established and the corresponding probabilistic description provides a common language that supports simple algorithmic proposals. The presentation was focused on coupled thermal transfers for their very wide application significance, but the illustrated framework is nonetheless much wider than this particular disciplinary field and other linear coupled physics can be studied similarly (see for instance [120–122]). Of course, many questions about coupled path spaces are still widely open, which were left out from the present article. Several of these questions are getting addressed in ongoing works that already started to provide insightful perspectives, as for instance:

- computing spatial gradients or parametric and geometric sensitivities; solutions are starting to emerge from a better understanding of the information carried by thermal paths that is available for further quantitative analysis [11, 123–126];

- the question of non-linearly coupled physics; several theoretical advances in the domain may lead to practical solutions for probe computations, which preserve the essential properties of the present methodology [103, 122, 127–129].

- in a very general way, revisiting physical intuitions using path integrals that include coupling has consequences in terms of analysis because of the possibility to read the structure of the coupling inside a single trajectory space.

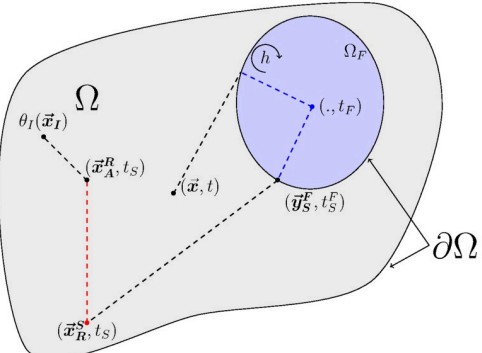

(a) Green functions method

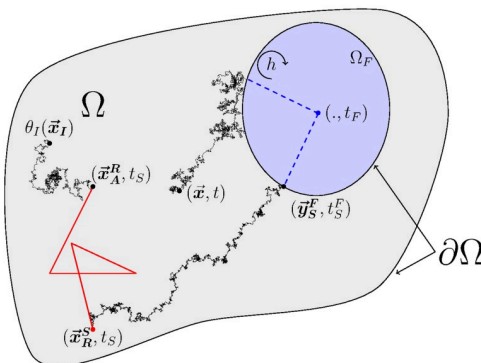

(b) Stochastic process method

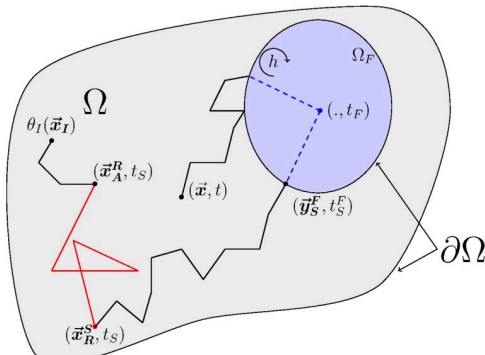

(c) Stochastic process method

**Fig 9. Fig 7 is completed by the representation of a diffusive random walk under the approximation of a $\delta$-sphere random walk.** (a) Green functions method, (b) Stochastic process method, and (c) Stochastic process method.

## APPENDIX

### A Radiation linearized in temperature

We start from the stationary radiative transfer equation formulated in terms of monochromatic specific intensity $I_\nu \equiv I_\nu(\vec{x}, \vec{u}, t)$ at position $\vec{x}$, in direction $\vec{u}$ at time $t$ and frequency $\nu$:

$$\vec{u} . \vec{\nabla} I_\nu = -k_a^\nu I_\nu + k_a^\nu I_\nu^{eq} - k_s^\nu I_\nu + k_s^\nu \int_{4\pi} p_s^\nu(\vec{u}|\vec{u}') I_\nu' \mathrm{d}\vec{u}' \tag{64}$$

where $k_a^\nu$ is the absorption coefficient, $k_s^\nu$ the scattering coefficient, $p_s^\nu$ the scattering phase function, $I_\nu' \equiv I_\nu(\vec{x}, \vec{u}', t)$ and $I_\nu^{eq} \equiv I_\nu^{eq}(\theta(\vec{x}, t))$ the specific equilibrium intensity at temperature $\theta(\vec{x}, t)$ of the solid or fluid. Although the radiative transfer is stationary, $I_\nu$ depends on $t$ due to the evolution of the solid/fluid temperature.

Assuming that at all times and positions, the solid/fluid temperature remains close to a reference temperature $\theta_{\mathrm{ref}}$, the temperature dependence of the specific equilibrium intensity can be linearized:

$$I_\nu^{eq}(\theta) \approx I_\nu^{eq}(\theta_{\mathrm{ref}}) + \partial_\theta I_\nu^{eq}(\theta_{\mathrm{ref}})(\theta - \theta_{\mathrm{ref}}) \tag{65}$$

Equilibrium properties allow to write:

$$0 = -k_a^\nu I_\nu^{eq} + k_a^\nu I_\nu^{eq} - k_s^\nu I_\nu^{eq} + k_s^\nu \int_{4\pi} p_s^\nu(\vec{u}|\vec{u}') I_\nu^{eq} \mathrm{d}\vec{u}' \tag{66}$$

Introducing the notation $\tilde{I}_\nu = I_\nu - I_\nu^{eq}(\theta_{\mathrm{ref}})$ for the perturbations and subtracting Eqs (64) and (66), the radiative transfer equation under the assumption (65) can be written as follows:

$$\vec{u} . \vec{\nabla} \tilde{I}_\nu \approx -k_a^\nu \tilde{I}_\nu + k_a^\nu \partial_\theta I_\nu^{eq}(\theta_{ref})(\theta - \theta_{ref}) - k_s^\nu \tilde{I}_\nu + k_s^\nu \int_{4\pi} p_s^\nu(\vec{u}|\vec{u}') \tilde{I}_\nu' \mathrm{d}\vec{u}' \tag{67}$$

We choose to rewrite this equation using the radiance temperature $\theta_{R,\vec{u}}^\nu$ in the direction $\vec{u}$. This radiance temperature is a spectral and directional quantity defined as the temperature for which the equilibrium specific intensity is equal to the specific intensity:

$$I_\nu^{eq}(\theta_{R,\vec{u}}^\nu(\vec{x}, t)) = I_\nu(\vec{x}, \vec{u}, t) \tag{68}$$

Using Eq (65),

$$I_\nu \approx I_\nu^{eq}(\theta_{\mathrm{ref}}) + \partial_\theta I_\nu^{eq}(\theta_{\mathrm{ref}})(\theta_{R,\vec{u}}^\nu - \theta_{\mathrm{ref}}) \tag{69}$$

and therefore,

$$\tilde{I}_\nu \approx \partial_\theta I_\nu^{eq}(\theta_{\mathrm{ref}})(\theta_{R,\vec{u}}^\nu - \theta_{\mathrm{ref}}) \tag{70}$$

Eq (67) becomes:

$$\boxed{\vec{u} . \vec{\nabla} \theta_{R,\vec{u}}^\nu \approx -k_a^\nu \theta_{R,\vec{u}}^\nu + k_a^\nu \theta - k_s^\nu \theta_{R,\vec{u}}^\nu + k_s^\nu \int_{4\pi} p_s^\nu(\vec{u}|\vec{u}') \theta_{R,\vec{u}'}^\nu \mathrm{d}\vec{u}'} \tag{71}$$

In the energy conservation equation, the radiation balance term $\psi_R$ is defined as the difference between the absorbed and emitted power densities, $\psi_R = \psi_{absorbed} - \psi_{emitted}$ with

$$\psi_{absorbed} = \int_{4\pi} \mathrm{d}\vec{u} \int_0^{+\infty} \mathrm{d}\nu \ k_a^\nu I_\nu \tag{72}$$

and

$$\psi_{emitted} = \int_{4\pi} d\vec{u} \int_0^{+\infty} dv \ k_a^v I_v^{eq} \tag{73}$$

Using the Stefan-Boltzmann law,

$$\int_0^{+\infty} I_v^{eq}(\theta_{\text{ref}}) dv = \frac{\sigma \theta_{\text{ref}}^4}{\pi}$$

and the previous assumptions, we can write:

$$
\begin{aligned}
\psi_R &= \int_{4\pi} d\vec{u} \int_0^{+\infty} dv k_a^v \partial_\theta I_v^{eq}(\theta_{\text{ref}})(\theta_{R,\vec{u}}^v - \theta) \\
&= \int_{4\pi} d\vec{u} \int_0^{+\infty} dv k_a^v \partial_\theta I_v^{eq}(\theta_{\text{ref}})\theta_{R,\vec{u}}^v - \left( \int_{4\pi} d\vec{u} \int_0^{+\infty} dv k_a^v \partial_\theta I_v^{eq}(\theta_{\text{ref}}) \right)\theta \\
&= 16 k_a \sigma \theta_{\text{ref}}^3 \left( \left( \int_{4\pi} \frac{1}{4\pi} d\vec{u} \int_0^{+\infty} dv p_N(v)\theta_{R,\vec{u}}^v \right) - \theta \right)
\end{aligned}
\tag{74}
$$

with

$$p_N(v) = \frac{k_a^v}{k_a} p_k(v) \tag{75}$$

where

$$p_k(v) = \frac{\pi \partial_\theta I_v^{eq}(\theta_{\text{ref}})}{4\sigma \theta_{\text{ref}}^3} \tag{76}$$

and

$$k_a = \int_0^{+\infty} p_k(v) dv \ k_a^v \tag{77}$$

We finally retain:

$$
\begin{cases}
\psi_R &= \zeta(\theta_R - \theta) \\
\theta_R &= \int_{4\pi} \frac{1}{4\pi} d\vec{u} \int_0^{+\infty} dv p_N(v)\theta_{R,\vec{u}}^v
\end{cases}
\tag{78}
$$

with $\zeta = 16 k_a \sigma \theta_{\text{ref}}^3$

## B Definitions for $\Theta_{F_i}(t)$

Eq (5), which is a first order time equation for the fluid temperature variable, fits perfectly into the illustrative case described in Fig 2 and can be reformulated as:

$$
\begin{cases}
\dfrac{d\theta_{F_i}}{dt}(t) = -\alpha_{F_i}\left(\theta_{F_i}(t) - \theta_{F_i}^*(t)\right) \\[4mm]
\alpha_{F_i} = \dfrac{\zeta_i \mathcal{V}_{F_i} + \int_{\partial\Omega_{F_i}} h_F(\vec{\boldsymbol{y}}_{\boldsymbol{s}})\mathrm{d}\vec{\boldsymbol{y}}_{\boldsymbol{s}}}{\rho_i C_i \mathcal{V}_{F_i}} \\[4mm]
\theta_{F_i}^*(t) = \dfrac{\zeta_i \int_{\Omega_{F_i}} \theta_R(\vec{\boldsymbol{x}}_{\boldsymbol{R}},t)\mathrm{d}\vec{\boldsymbol{x}}_{\boldsymbol{R}} + \int_{\partial\Omega_{F_i}} h_F(\vec{\boldsymbol{y}}_{\boldsymbol{s}})\theta_S(\vec{\boldsymbol{y}}_{\boldsymbol{s}},t)\mathrm{d}\vec{\boldsymbol{y}}_{\boldsymbol{s}}}{\zeta_i \mathcal{V}_{F_i} + \int_{\partial\Omega_{F_i}} h_F(\vec{\boldsymbol{y}}_{\boldsymbol{s}})\mathrm{d}\vec{\boldsymbol{y}}_{\boldsymbol{s}}} \\[4mm]
\theta_{F_i}(t_I) = \theta_I
\end{cases}
\tag{79}
$$

The development then leads to the same probabilistic formulation as in Fig 2, with $\tilde{\Theta}_{F_i}$ the random variable whose expectation is $\theta_{F_i}$:

$$
\tilde{\Theta}_{F_i}(t) \;\; = \;\; \mathcal{B}(p_I^{F_i})\theta_I + \left(1 - \mathcal{B}(p_I^{F_i})\right)\theta_{F_i}^*(T^{F_i})
\tag{80}
$$

$$
\theta_{F_i}(t) \;\; = \;\; \mathbb{E}\left[\tilde{\Theta}_{F_i}(t)\right]
\tag{81}
$$

where:

- $p_I^{F_i} = \exp\left(-\alpha_{F_i}(t - t_I)\right)$; with $t_I$ the initial time.

- $\mathcal{B}(p)$ is a Bernoulli r.v. with parameter $p$

- $T^{F_i}$ is a r.v with an exponential distribution of parameter $\alpha_{F_i}$

Meanwhile, expressions provided by 80 and 81 are not fully satisfactory because the source term $\theta_{F_i}^*(t)$ involves integrals of both temperatures $\theta_R$ and $\theta_S$. It is thus necessary to carry out the probabilization of $\theta_{F_i}^*(t)$ by writing the temperature of the fluid as the expectation of a random variable which takes either the value of $\theta_R$ or that of $\theta_S$ (or $\theta_I$).

Such an expression will be compatible with the final objective, which remains the coupling between submodels. Indeed, the temperatures $\theta_R$ and $\theta_S$, which are prescribed here, will be the coupling variables when solving the whole model in Eq (6).

Let us define the area $\mathcal{S}_{F_i}$ of the domain $\partial\Omega_{F_i}$ and $\bar{h}_{F_i} = \int_{\partial\Omega_{F_i}} h_F(\vec{\boldsymbol{y}}_{\boldsymbol{s}})\mathrm{d}\vec{\boldsymbol{y}}_{\boldsymbol{s}}/\mathcal{S}_{F_i}$ the convection coefficient $h_F$ averaged over the boundary. Hence:

$$
\theta_{F_i}^*(t) = \frac{\zeta_i \mathcal{V}_{F_i}}{\zeta_i \mathcal{V}_{F_i} + \bar{h}_{F_i}\mathcal{S}_{F_i}} \int_{\Omega_{F_i}} \frac{1}{\mathcal{V}_{F_i}}\theta_R(\vec{\boldsymbol{x}}_{\boldsymbol{R}},t)\mathrm{d}\vec{\boldsymbol{x}}_{\boldsymbol{R}} + \frac{\bar{h}_{F_i}\mathcal{S}_{F_i}}{\zeta_i \mathcal{V}_{F_i} + \bar{h}_{F_i}\mathcal{S}_{F_i}} \int_{\partial\Omega_{F_i}} \frac{h_F(\vec{\boldsymbol{y}}_{\boldsymbol{s}})}{\bar{h}_{F_i}\mathcal{S}_{F_i}}\theta_S(\vec{\boldsymbol{y}}_{\boldsymbol{s}},t)\mathrm{d}\vec{\boldsymbol{y}}_{\boldsymbol{s}}
$$

Which finally leads to the expression of the fluid temperature (Eq (15)):

$$\theta_{F_i}(t) = \quad g_{F_i,I}(t|t_I)\theta_I$$

$$+ \int_{t_I}^{t} \int_{\partial\Omega_{F_i}} g_{F_i,S}(t|\vec{y}_S,\tau)\theta_S(\vec{y}_S,t)\mathrm{d}\vec{y}_S d\tau$$

$$+ \int_{t_I}^{t} \int_{\Omega_{F_i}} g_{F_i,R}(t|\vec{x}_R,\tau)\theta_R(\vec{x}_R,\tau)\mathrm{d}\vec{x}_R d\tau$$

where

- $g_{F_i,I}(t|t_I) = p_I^{F_i}$

- $g_{F_i,S}(t|\vec{y}_S,\tau) = (1 - p_R^{F_i})\frac{h(\vec{y}_S)}{h\mathcal{S}_{F_i}}\alpha_{F_i}\exp\left(-\alpha_{F_i}(t-\tau)\right)$

- $g_{F_i,R}(t|\vec{x}_R,\tau) = p_R^{F_i}\frac{1}{\mathcal{V}_{F_i}}\alpha_{F_i}\exp\left(-\alpha_{F_i}(t-\tau)\right)$

  with $p_R^{F_i} = \frac{\zeta_i\mathcal{V}_{F_i}}{\zeta_i\mathcal{V}_{F_i}+\bar{h}_{F_i}\mathcal{S}_{F_i}}$.

  Defining the two independent variables

- $\vec{Y}_S^{F_i}$ a random position variable following the distribution $\frac{h_F(\vec{y}_S)}{h_{F_i}\mathcal{S}_{F_i}}$ on the surface $\partial\Omega_{F_i}$

- $\vec{X}_R^{F_i}$ a random position variable following the uniform distribution $\frac{1}{\mathcal{V}_{F_i}}$ on $\Omega_{F_i}$

  we can write Eqs (16) and (17) which define the temperature of the fluid as an expectation:

$$\theta_{F_i}(t) = \mathbb{E}[\Theta_{F_i}(t)]$$

with

$$\Theta_{F_i}(t) = \quad \mathcal{B}_1(p_I^{F_i})\theta_I$$

$$+ \left(1 - \mathcal{B}_1(p_I^{F_i})\right)\left\{\mathcal{B}_2(p_R^{F_i})\theta_R(\vec{X}_R^{F_i}, T^{F_i}) + \left(1 - \mathcal{B}_2(p_R^{F_i})\right)\theta_S(\vec{Y}_S^{F_i}, T^{F_i})\right\}$$

When dealing with coupling, we will then work, for each fluid subvolume $\Omega_{F_i}$, with the random variables $\Theta_{F_i}(t)$ rather than $\tilde{\Theta}_{F_i}(t)$.

## C Definitions for $\Theta_S(\vec{x}, t)$

This appendix aims at providing the definitions of the random variables and probabilities that appear in expression 21, reported here for the sake of clarity:

$$\Theta_S(\vec{x}, t) = \quad \mathcal{B}_1(p_I^S)\theta_I(\vec{X}_I^S) + (1 - \mathcal{B}_1(p_I^S))\mathcal{B}_2(p_2^S)\theta_R(\vec{X}_R^S, T_R^S)$$

$$+ (1 - \mathcal{B}_1(p_I^S))(1 - \mathcal{B}_2(p_2^S))\mathcal{B}_3(p_3^S)\theta_F(\vec{Y}_F^S, T_F^S)$$

$$+ (1 - \mathcal{B}_1(p_I^S))(1 - \mathcal{B}_2(p_2^S))(1 - \mathcal{B}_3(p_3^S))\theta_D(\vec{Y}_D^S, T_D^S)$$

- Definition of the probabilities:

$$
\begin{aligned}
p_I^S(\vec{x}, t|t_I) &= \int_{\Omega_S} g_{S,I}(\vec{x}, t|\vec{x}_I, t_I)\mathrm{d}\vec{x}_I \\
p_R^S(\vec{x}, t|t_I) &= \int_{t_I}^t \int_{\Omega_S} g_{S,R}(\vec{x}, t|\vec{x}_R, t_R)\mathrm{d}\vec{x}_R\mathrm{d}t_R \\
p_{\partial\Omega_S^D}^S(\vec{x}, t|t_I) &= \int_{t_I}^t \int_{\partial\Omega_S^D} g_{S,\partial\Omega_S^D}(\vec{x}, t|\vec{y}_F, t_F)\mathrm{d}\vec{y}_F\mathrm{d}t_F \\
p_{\partial\Omega_D}^S(\vec{x}, t|t_I) &= \int_{t_I}^t \int_{\partial\Omega_D} g_{S,\partial\Omega_D}(\vec{x}, t|\vec{y}_D, t_D)\mathrm{d}\vec{y}_D\mathrm{d}t_D
\end{aligned}
$$

with the relation (under the restriction conditions defined in Eq (8))

$$
p_I^S + p_R^S + p_{\partial\Omega_S^D}^S + p_{\partial\Omega_D}^S = 1
$$

so that the following quantities can be considered as probabilities:

$$
p_2^S = \frac{p_R^S}{1 - p_I^S} \quad \text{and} \quad p_3^S = \frac{p_{\partial\Omega_S^D}^S}{1 - p_I^S - p_R^S}
$$

- Definition of the probability density functions:

$$
\begin{aligned}
p_{\vec{X}_I^S}(\vec{x}, t|\vec{x}_I, t_I) &= g_{S,I}(\vec{x}, t|\vec{x}_I, t_I)/p_I^S(\vec{x}, t|t_I) \\
p_{(\vec{X}_R^S, T_R^S)}(\vec{x}, t|\vec{x}_R, t_R) &= g_{S,R}(\vec{x}, t|\vec{x}_R, t_R)/p_R^S(\vec{x}, t|t_I) \\
p_{(\vec{Y}_F^S, T_F^S)}(\vec{x}, t|\vec{y}_F, t_F) &= g_{S,\partial\Omega_S^D}(\vec{x}, t|\vec{y}_F, t_F)/p_{\partial\Omega_S^D}^S(\vec{x}, t|t_I) \\
p_{(\vec{Y}_D^S, T_D^S)}(\vec{x}, t|\vec{y}_D, t_D) &= g_{S,\partial\Omega_D}(\vec{x}, t|\vec{y}_D, t_D)/p_{\partial\Omega_D}^S(\vec{x}, t|t_I)
\end{aligned}
$$

- Definition of the random variables:

  - $\mathcal{B}(p)$ is a Bernoulli r.v. with parameter $p$

  - $\vec{X}_I^S$ is a r.v. with distribution $p_{\vec{X}_I^S}$

  - $(\vec{X}_R^S, T_R^S)$ is a paired r.v with distribution $p_{(\vec{X}_R^S, T_R^S)}$

  - $(\vec{Y}_F^S, T_F^S)$ is a paired r.v with distribution $p_{(\vec{Y}_F^S, T_F^S)}$

  - $(\vec{Y}_D^S, T_D^S)$ is a paired r.v with distribution $p_{(\vec{Y}_D^S, T_D^S)}$

which are independent from each others.

## D Definitions for $\Theta_{R,\vec{u}}(\vec{x}, t)$

This appendix aims at providing the definitions of the random variables and probabilities present in expression 25, reported here for the sake of clarity:

$$\Theta_{R,\vec{u}}(\vec{x}, t) = \mathcal{B}(p_A^R(\vec{x}, \vec{u}))\theta(\vec{X}_A^R, t) + (1 - \mathcal{B}(p_A^R(\vec{x}, \vec{u})))\theta_{R,\partial\Omega_R,\vec{u}_R}(\vec{Y}_R^R, t)$$

- Definition of the probabilities:

$$p_A^R(\vec{x}, \vec{u}) = \int_\Omega \int_{\mathbb{S}^2} g_{R,A}(\vec{x}, \vec{u}|\vec{x}', \vec{u}')\mathrm{d}\vec{u}'\mathrm{d}\vec{x}'$$

$$p_R^R(\vec{x}, \vec{u}) = \int_{\partial\Omega_R} \int_{\mathbb{S}_+^2} g_{R,\partial\Omega_R}(\vec{x}, \vec{u}|\vec{y}, \vec{u}')\mathrm{d}\vec{u}'\mathrm{d}\vec{y}$$

with the relation (under the restriction conditions defined in Eq (8)):

$$p_A^R + p_R^R = 1$$

- Definition of the probability density functions:

$$p_{\vec{X}_A^R}(\vec{x}, \vec{u}|\vec{x}_A) = \left( \int_{\mathbb{S}^2} g_{R,A}(\vec{x}, \vec{u}|\vec{x}_A, \vec{u}_A)\mathrm{d}\vec{u}_A \right)/p_A^R(\vec{x}, \vec{u})$$

$$p_{(\vec{Y}_R^R, \vec{u}_R)}(\vec{x}, \vec{u}|\vec{y}_R, \vec{u}_R) = g_{R,\partial\Omega_R}(\vec{x}, \vec{u}|\vec{y}_R, \vec{u}_R)/p_R^R(\vec{x}, \vec{u})$$

- Definition of the random variables:

  - $\mathcal{B}(p)$ is a r.v with parameter $p$

  - $\vec{X}_A^R$ is the r.v with distribution $p_{\vec{X}_A^R}$

  - $(\vec{Y}_R^R, \vec{U}_R)$ is a paired r.v. with distribution $p_{(\vec{Y}_R^R, \vec{U}_R)}$

which are independent from each others.

## E Random Walk on Sphere and equivalent

For sampling Brownian motion in a confined environment, there are almost only approximate methods that use numerical parameters on which the accuracy of the method depends [38, 80]. Among the most popular applications, it is worth mentioning the elegance of so-called "first passage of a trajectory over a fictitious boundary" methods [130, 131].

The most common approach when it comes to sampling contributions according to the conductive Green function, in a close domain $\Omega$ with a Dirichlet boundary condition, is the Walk on Sphere proposition [28, 31, 118, 119]. It consists in the random sampling of a point (both in space and time) over successive spheres of maximal radius, centered on the current position, as illustrated in Fig 10(a) (spheres are tangent to the boundary of the domain, and entirely fit inside the domain). From the first passage Green function over a sphere, it is possible to deduce:

1. a distribution of exit times,

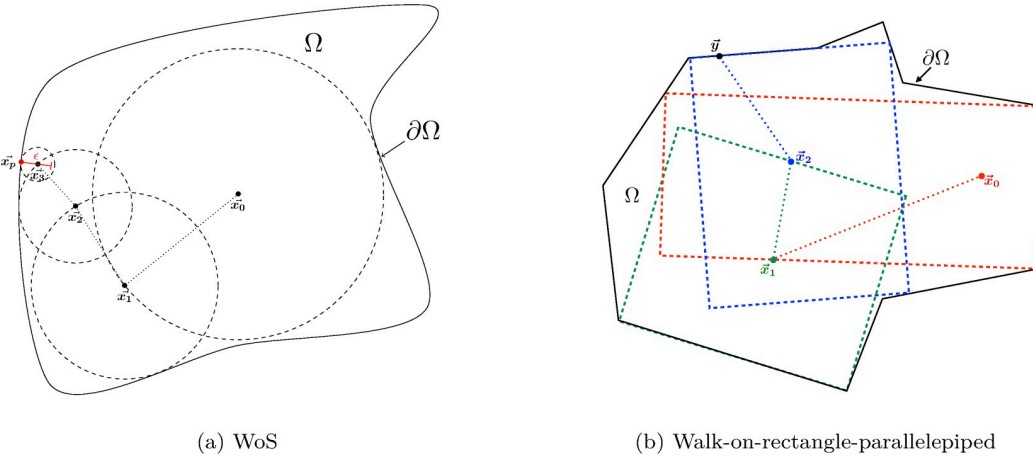

(a) WoS (b) Walk-on-rectangle-parallelepiped

**Fig 10. Illustration of methods based on Green's function first-passage algorithms.** (a) WoS and (b) Walk-on-rectangle-parallelepiped.

2. a distribution of exit positions over the sphere (that follows a uniform law when conductivity $\lambda$ is uniform!)

3. a distribution of positions inside the sphere, given that information is confined to the inside of the sphere during the time elapsed from the initial condition.

The numerical strategy in order to find a exit position over the boundary of the domain consists in setting an arbitrary thickness $\epsilon$ to the boundary. As a consequence, a position is considered to have reached the boundary as soon as its distance to the boundary is less than $\epsilon$ (this is necessary, since the contact between the sphere and $\Omega$ is most of the time reduced to a point).

With this approximation and using the previously described three distributions, the contribution associated with the current sphere can be sampled. In this case, three situations must be considered:

- either the initial condition has been reached inside the sphere, and the MC weight is the initial temperature for the corresponding position, sampled using the distribution of point 3.,

- or a position is reached at the surface of the sphere located at a distance smaller than $\epsilon$ from the boundary, in which case the MC weight is the known temperature at the boundary,

- or a position is reached at the surface of the sphere located at a distance greater than $\epsilon$ from the boundary (at a time that is per construction closer to the initial condition), in which case recursivity occurs.

A "walk" over spheres therefore emerges, until a boundary condition is reached. First, thickening the boundary using parameter $\epsilon$ is a source of uncertainty. This approximation is not, in fact, an issue, because the average number of jumps that is required to reach the boundary increases as $|\log(\epsilon)|$ [31, 130]; it is therefore possible to imagine a value of $\epsilon$ that is compatible with the numerical accuracy inherent to the representation of numbers, which is the limiting approximation. Secondly, the implementation needs, for each jump, to solve an optimization problem: identify the sphere of maximal radius contained in $\Omega$ and centered on the current position. This optimization problem may be computationally expensive for a high

level of geometric complexity. Current research on this topic has already been translated in more efficient intersection libraries [132, 133].

In the same line of thought, recent advances [34, 35, 76, 77] propose an effcient methodology in order to sample contributions in a general polyhedral domain with a Dirichlet boundary condition, as illustrated by Fig 10(b) (the strategy is similar to the Walk on Sphere, but with parallelepipedic rectangles). This proposition consists in computing the Green function within a parallelepipedic rectangle of arbitrary dimensions, in order to obtain:

1. a distribution of exit times (that depends on the probe position)

2. a distribution of exit positions (that depends on the probe position and the exit time)

3. a distribution of positions inside the rectangle, knowing that the information remains confined to the inside of the rectangle domain during the time period from the initial condition (that depends on the probe position and the observation time)

The proposition then consists in a practical and ingenious process to generate parallelepipedic rectangles contained in the polyhedron and containing the probe position, while maximizing the contact area between this rectangle and the boundary of the polyhedron. It is then possible, from the three distributions, to sample a contribution associated with the rectangle. In each case, three situations have to be examined:

- either the initial condition inside the rectangle is reached, and the MC weight is the initial temperature at the corresponding position,

- or a position is reached at the portion of the surface of the rectangle that is shared with the surface of the polyhedron, and the MC weight is the imposed boundary temperature,

- or a position is reached at the portion of the surface of the rectangle that is inside the polyhedron (at a time closer to the initial condition), in which case recursivity occurs.

A "walk" on rectangles emerges, until a limit condition is reached. Contrary to the Walk on Sphere, there is no thickening parameter, making this approach a reference method since the various contributions are sampled in a exact way. One of the remaining questions, that is currently not addressed on a theoretical point of view, is the capacity to generate such parallelepipedic rectangles efficiently enough in a complex geometry. Nonetheless, even if it was possible to produce these volumes in an optimal way, accounting for couplings and heterogeneous parameters is a consequent additional work that remains to be conducted.

Attempts to address this question were made, notably by trying to sample contributions in a domain with non-uniform conductivity, as for instance in the case of a discontinuity at an interface [36, 74, 78, 79], or with Robin or Neumann boundary conditions [104, 134, 135].

## F Consistency of the approximate $\delta$-sphere random walk on infinite domains

In order to investigate the consistency of the approximate model and the convergence speed of $\tilde{\theta}_S$ towards $\theta_S$ as a function of $\delta$, a solution to the coupled conducto-radiative tridimensional model provided by Eqs (51) and (52) will be proposed, while specifying the radiative physics of interest.

Eq (52) can be seen as the solution of a radiative transfer equation that is obtained after describing the underlying process. A rather simple way of obtaining this formulation consists in reformulating the differential radiative transfer equation under its integral form. In the considered radiative physics, the collisional term is associated to absorption and scattering

processes for spatially uniform radiative properties. For the needs of the computation presented in this appendix, the phase function is chosen isotropic. The radiative transfer equation under its integral form can then be written as:

$$\theta_R(\vec{x}, t) = \int_{\mathbb{S}^2} \frac{1}{4\pi} d\vec{u} \int_0^{+\infty} k_e e^{-k_e l} dl \left( \frac{k_a}{k_e} \theta_S + \frac{k_s}{k_e} \theta_R \right)_{|(\vec{x}+\vec{u}l, t)} \quad , (t, \vec{x}) \in ]t_I, +\infty[ \times \mathbb{R}^3 \quad (82)$$

Note that using the iterated kernel procedure over the Fredholm Eq (82), the solution appears as a development of a Neumann series, which formally defines the integration path space as described by Eq (52). This additionnal step is not useful for the needs of this appendix.

Using Eq (82), $\theta_S$ and $\tilde{\theta}_S$ are respectively the solutions of coupled Eqs (83) and (84). For the original model,

$$\begin{cases} \rho C \partial_t \theta_S = \lambda \Delta \theta_S + \zeta(\theta_R - \theta_S) & , (t, \vec{x}) \in ]t_I, +\infty[ \times \mathbb{R}^3 \\[2mm] \theta_S(\vec{x}, t_I) = \theta_I(\vec{x}) & , \vec{x} \in \mathbb{R}^3 \\[2mm] \theta_R(\vec{x}, t) = \int_{\mathbb{S}^2} \frac{1}{4\pi} d\vec{u} \int_0^{+\infty} k_e e^{-k_e l} dl \left( \frac{k_a}{k_e} \theta_S + \frac{k_s}{k_e} \theta_R \right)_{|(\vec{x}+\vec{u}l, t)} & , (t, \vec{x}) \in ]t_I, +\infty[ \times \mathbb{R}^3 \end{cases} \quad (83)$$

and for the approximate model,

$$\begin{cases} \rho C \partial_t \tilde{\theta}_S = \lambda \left( \dfrac{6 \int_{\mathbb{S}^2} \frac{1}{4\pi} d\vec{u} \tilde{\theta}_{S,\vec{u}} - 6\tilde{\theta}_S}{\delta^2} \right) + \zeta(\tilde{\theta}_R - \tilde{\theta}_S) & , (t, \vec{x}) \in ]t_I, +\infty[ \times \mathbb{R}^3 \\[4mm] \tilde{\theta}_S(\vec{x}, t_I) = \theta_I(\vec{x}) & , \vec{x} \in \mathbb{R}^3 \\[4mm] \tilde{\theta}_R(\vec{x}, t) = \int_{\mathbb{S}^2} \frac{1}{4\pi} d\vec{u} \int_0^{+\infty} k_e e^{-k_e l} dl \left( \frac{k_a}{k_e} \tilde{\theta}_S + \frac{k_s}{k_e} \tilde{\theta}_R \right)_{|(\vec{x}+\vec{u}l, t)} & , (t, \vec{x}) \in ]t_I, +\infty[ \times \mathbb{R}^3 \end{cases} \quad (84)$$

where $k_a$, $k_s$ and $k_e = k_a + k_s$ are respectively the absorption, scattering and extinction coefficients. Let us recall that $\tilde{\theta}_{S,\vec{u}} \equiv \tilde{\theta}_S(\vec{x} + \delta\vec{u}, t)$ and that $\mathbb{S}^2$ is the unit sphere of dimension two.

## Fourier expansion

$\tilde{\theta}_S(\vec{x}, t)$ is obtained by a development over a Fourier basis, in order to get the dispersion relation for the approximate model. Notation $\hat{\tilde{\theta}}_S(\vec{k}, t)$ is used for the component on the basis associated with the wave vector $\vec{k}$ (Fourier transform).

$$\rho C \partial_t \left( \int_{\mathbb{R}^3} \hat{\tilde{\theta}}_S(\vec{k}, t) e^{i\vec{k}.\vec{x}} d\vec{k} \right) = \begin{pmatrix} \dfrac{6\lambda}{\delta^2} \int_{\mathbb{S}^2} \frac{1}{4\pi} du \int_{\mathbb{R}^3} \hat{\tilde{\theta}}_S(\vec{k}, t) e^{i\vec{k}.(\vec{x}+\delta\vec{u})} d\vec{k} \\[3mm] - \dfrac{6\lambda}{\delta^2} \int_{\mathbb{R}^3} \hat{\tilde{\theta}}_S(\vec{k}, t) e^{i\vec{k}.\vec{x}} d\vec{k} \\[3mm] + \zeta \left( \int_{\mathbb{R}^3} \hat{\tilde{\theta}}_R(\vec{k}, t) e^{i\vec{k}.\vec{x}} d\vec{k} - \int_{\mathbb{R}^3} \hat{\tilde{\theta}}_S(\vec{k}, t) e^{i\vec{k}.\vec{x}} d\vec{k} \right) \end{pmatrix} \quad (85)$$

that is reformulated as follows,

$$\int_{\mathbb{R}^3} \rho C \partial_t \hat{\tilde{\theta}}_S(\vec{k},t) e^{i\vec{k}.\vec{x}} d\vec{k} = \int_{\mathbb{R}^3} \left( \begin{array}{c} \frac{6\lambda}{\delta^2} \left( \int_{\mathbb{S}^2} \frac{1}{4\pi} d\vec{u} e^{i\delta\vec{k}.\vec{u}} - 1 \right) \hat{\tilde{\theta}}_S(\vec{k},t) \\ + \quad \zeta(\hat{\tilde{\theta}}_R - \hat{\tilde{\theta}}_S) \end{array} \right) e^{i\vec{k}.\vec{x}} d\vec{k} \qquad (86)$$

Using the fact that $\{\vec{x} \mapsto e^{i\vec{k}.\vec{x}}\}_{\vec{k} \in \mathbb{R}^3}$ is a Hilbert basis, a relation for each component of the associated wave vector $\vec{k}$ is obtained:

$$\partial_t \hat{\tilde{\theta}}_S = \frac{6D}{\delta^2} \left( \int_{\mathbb{S}^2} \frac{1}{4\pi} d\vec{u} \ e^{i\delta\vec{k}.\vec{u}} - 1 \right) \hat{\tilde{\theta}}_S + \frac{\zeta}{\rho C} \left( \hat{\tilde{\theta}}_R - \hat{\tilde{\theta}}_S \right)$$

$$\partial_t \hat{\tilde{\theta}}_S = \frac{6D}{\delta^2} \left( \int_{\mathbb{S}^2_+} \frac{1}{2\pi} d\vec{u} \ \frac{e^{i\delta\vec{k}.\vec{u}} + e^{-i\delta\vec{k}.\vec{u}}}{2} - 1 \right) \hat{\tilde{\theta}}_S + \frac{\zeta}{\rho C} \left( \hat{\tilde{\theta}}_R - \hat{\tilde{\theta}}_S \right)$$

$$\partial_t \hat{\tilde{\theta}}_S = \frac{6D}{\delta^2} \left( \int_0^{2\pi} \frac{1}{2\pi} d\phi \int_0^{\pi/2} \sin(\theta) d\theta \ \cos(\delta||\vec{k}||\cos(\theta)) - 1 \right) \hat{\tilde{\theta}}_S + \frac{\zeta}{\rho C} \left( \hat{\tilde{\theta}}_R - \hat{\tilde{\theta}}_S \right)$$

$$\partial_t \hat{\tilde{\theta}}_S = \frac{6D}{\delta^2} \left( \frac{\sin(\delta||\vec{k}||)}{\delta||\vec{k}||} - 1 \right) \hat{\tilde{\theta}}_S + \frac{\zeta}{\rho C} \left( \hat{\tilde{\theta}}_R - \hat{\tilde{\theta}}_S \right)$$

$$(87)$$

where $D = \lambda/\rho C$ is the thermal diffusivity. Similarly, the Fourier transform for $\tilde{\theta}_R$ is straightforward:

$$\hat{\tilde{\theta}}_R = \frac{k_a \arctan\left( \frac{||\vec{k}||}{k_e} \right)}{||\vec{k}|| - k_s \arctan\left( \frac{||\vec{k}||}{k_e} \right)} \hat{\tilde{\theta}}_S \qquad (88)$$

Combining Eqs (87) and (88):

$$\partial_t \hat{\tilde{\theta}}_S = \left( \frac{6D}{\delta^2} \left( \frac{\sin(\delta||\vec{k}||)}{\delta||\vec{k}||} - 1 \right) \hat{\tilde{\theta}}_S + \frac{\zeta}{\rho C} \left( \frac{k_a \arctan\left( \frac{||\vec{k}||}{k_e} \right)}{||\vec{k}|| - k_s \arctan\left( \frac{||\vec{k}||}{k_e} \right)} - 1 \right) \right) \hat{\tilde{\theta}}_S \qquad (89)$$

from which it is deduced that:

$$\hat{\tilde{\theta}}_S(\vec{k},t) = \hat{\theta}_I(\vec{k},t) e^{-(t-t_I)/\tilde{\tau}} \qquad (90)$$

where the characteristic time $\tilde{\tau}$ associated to $\vec{k}$ verifies:

$$\tilde{\tau} = \left( -\frac{6D}{\delta^2} \left( \frac{\sin(\delta||\vec{k}||)}{\delta||\vec{k}||} - 1 \right) - \frac{\zeta}{\rho C} \left( \frac{k_a \arctan\left( \frac{||\vec{k}||}{k_e} \right)}{||\vec{k}|| + k_s \arctan\left( \frac{||\vec{k}||}{k_e} \right)} - 1 \right) \right)^{-1} \qquad (91)$$

Altogether, the analytical solution for $\tilde{\theta}_S$ is:

$$\tilde{\theta}_S(\vec{x}, t) = \int_{\mathbb{R}^3} \hat{\theta}_I(\vec{k}, t) e^{(t-t_I)/\tilde{\tau}} e^{i\vec{k}.\vec{x}} d\vec{k} \tag{92}$$

In order to perform a similar development over the $\theta_S(\vec{x}, t)$ model, the Fourier transform of the finite differences operator has to be replaced by the Fourier transform of the Laplacian operator:

$$\theta_S(\vec{x}, t) = \int_{\mathbb{R}^3} \hat{\theta}_I(\vec{k}, t) e^{-(t-t_I)/\tau} e^{i\vec{k}.\vec{x}} d\vec{k} \tag{93}$$

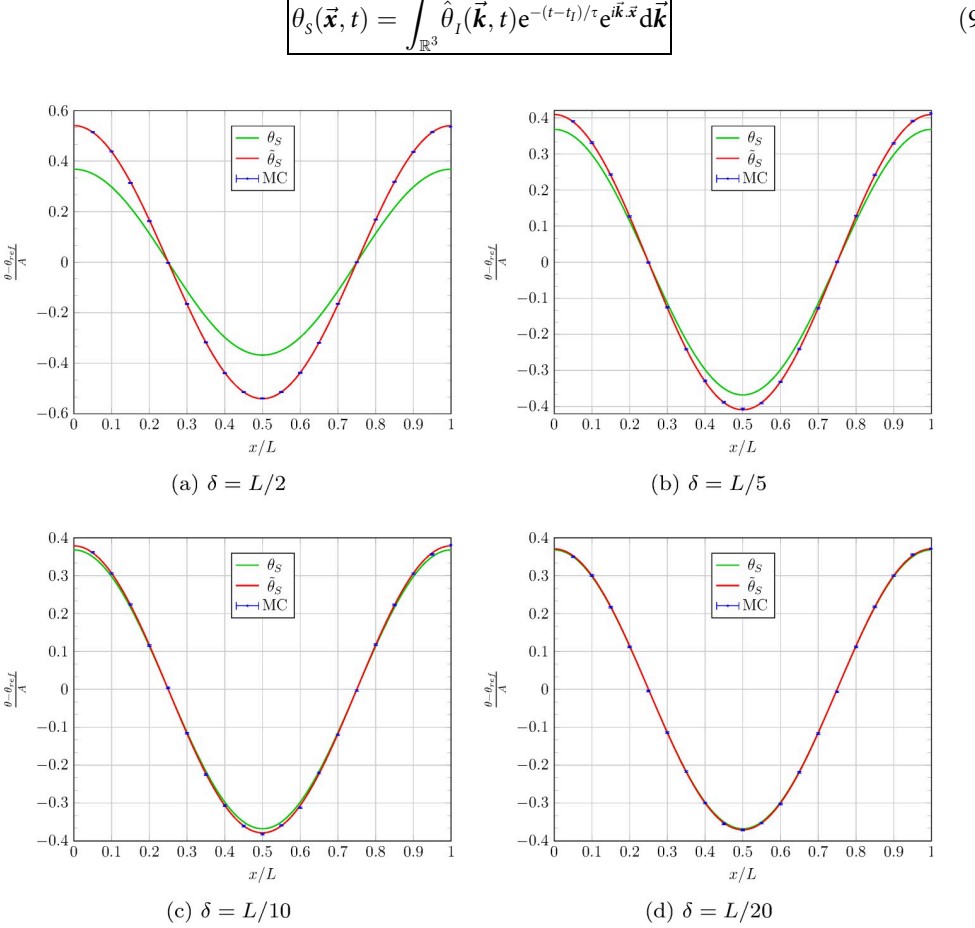

(a) $\delta = L/2$ (b) $\delta = L/5$

(c) $\delta = L/10$ (d) $\delta = L/20$

**Fig 11. Each $\theta_S$ and $\tilde{\theta}_S$ curve are the exact solutions of respectively (83) and (84).** Points that are denoted MC and associated errorbars are the solutions of a numerical resolution by MC over the approximate model. The initial temperature field is: $\theta_I(\vec{x}) = \theta_{\text{ref}} + A\cos(\vec{k}.\vec{x})$. Results have been obtained for a characteristic time $\tau$ and for position $x \in [0, L]$, $y = z = 0$ with $\vec{k} = (2\pi/L, 2\pi/L, 2\pi/L)$, for each figure. The only differentiating parameter for the four figures is the value of $\delta$ that is taken respectively in the $\{L/2, L/5, L/10 \text{ and } L/20\}$ set. The scattering radiative coefficient is null ($k_s = 0$) and the reference temperature $\theta_{\text{ref}}$ that is used in coefficient $\zeta$ is chosen so that a equivalent weight is given to conduction and radiation, through the constraint $D||\vec{k}||^2 = \frac{\zeta k_a}{\rho C ||\vec{k}||} \arctan\left(\frac{||\vec{k}||}{k_e}\right)$. (a) $\delta$ = L/2, (b) $\delta$ = L/5, (c) $\delta$ = L/10, and (d) $\delta$ = L/20.

where the characteristic time $\tau$ associated to $\vec{k}$ verifies:

$$\tau = \left( D||\vec{k}||^2 - \frac{\zeta}{\rho C}\left( \frac{k_a \arctan\left(\frac{||\vec{k}||}{k_e}\right)}{||\vec{k}|| + k_s \arctan\left(\frac{||\vec{k}||}{k_e}\right)} - 1 \right) \right)^{-1} \tag{94}$$

The consistency of the approximate model is now straightforward:

$$\tau = \lim_{\delta \mapsto 0} \tilde{\tau}$$

and the approximate model converges as $\delta^2$:

$$\frac{1}{\tau} - 1\tilde{\tau} = \frac{6D||\vec{k}||^4}{5!}\delta^2 + o(\delta^4) \tag{95}$$

The $\tilde{\theta}_S$ model approximation only finds its origin in the Laplacian associated to the diffusive process. The radiative part is evaluated exactly, which means that in parametric regions where radiation dominates over conduction, the approximation of the temperature field is nearly independent of the walk parameter $\delta$.

For illustration purposes, and without seeking completeness, Fig 11 shows how $\tilde{\theta}_S$ and $\theta_S$ temperature fields behave for a given set of parameters that correspond to radiative processes without scattering ($k_s = 0$) and equivalent weights for the conductive and radiative processes. It is worth mentioning that, without surprise, the MC computation behaves exactly as the approximate model whatever the value of the walk parameter $\delta$ (MC reconstructs Eq (92)). Furthermore, it should be noted that for a walking step equal to 1/20 of the characteristic length, the approximate model and the exact model agree within an outstanding level of accuracy.

## Supporting information

**S1 File. Nomenclature for all symbols used and abbreviations.**
(PDF)

## Author Contributions

**Conceptualization:** Jean Marc Tregan, Jean Luc Amestoy, Megane Bati, Jean-Jacques Bezian, Stéphane Blanco, Laurent Brunel, Cyril Caliot, Julien Charon, Jean-Francois Cornet, Christophe Coustet, Louis d'Alençon, Jeremi Dauchet, Sebastien Dutour, Simon Eibner, Mouna El Hafi, Vincent Eymet, Olivier Farges, Vincent Forest, Richard Fournier, Mathieu Galtier, Victor Gattepaille, Jacques Gautrais, Zili He, Frédéric Hourdin, Loris Ibarrart, Jean-Louis Joly, Paule Lapeyre, Pascal Lavieille, Marie-Helene Lecureux, Jacques Lluc, Marc Miscevic, Nada Mourtaday, Yaniss Nyffenegger-Péré, Lionel Pelissier, Lea Penazzi, Benjamin Piaud, Clément Rodrigues-Viguier, Gisele Roques, Maxime Roger, Thomas Saez, Guillaume Terrée, Najda Villefranque, Thomas Vourc'h, Daniel Yaacoub.

**Formal analysis:** Richard Fournier, Jacques Gautrais.

**Investigation:** Jean Marc Tregan, Louis d'Alençon, Jeremi Dauchet.

**Methodology:** Jean Marc Tregan, Stéphane Blanco, Jeremi Dauchet, Loris Ibarrart.

**Software:** Jean Marc Tregan, Christophe Coustet, Vincent Eymet, Vincent Forest, Richard Fournier, Loris Ibarrart, Benjamin Piaud, Clément Rodrigues-Viguier, Najda Villefranque.

**Supervision:** Stéphane Blanco.

**Validation:** Jean Marc Tregan, Stéphane Blanco, Christophe Coustet, Louis d'Alençon, Vincent Forest, Benjamin Piaud, Najda Villefranque.

**Writing – original draft:** Jean Marc Tregan, Stéphane Blanco, Jeremi Dauchet, Vincent Eymet, Richard Fournier, Jacques Gautrais, Najda Villefranque.

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
