## [Decision Letter · Decision Letter 0]

20 Jan 2023

PONE-D-22-29121Coupling radiative, conductive and convective heat-transfers in a single Monte Carlo algorithm: a general theoretical framework for linear situations.PLOS ONE

Dear Stéphane Blanco

Thank you for submitting your manuscript to PLOS ONE. After careful consideration, we feel that it has merit but does not fully meet PLOS ONE’s publication criteria as it currently stands. Therefore, we invite you to submit a revised version of the manuscript that addresses the points raised during the review process.

Reviewer 1's comments

The authors developed single Monte-Carlo algorithm for radiation, conduction and convection process. The authors used the theoretical frame works of propagators and Green's functions which demonstrate that a coupled model involving different physical phenomena can be probabilized. They extended the Feynman-Kac theory and stochastic processes. They addressed that the coupled Brownian trajectories compatible with the algorithmic design required by ray-tracing acceleration techniques in highly refined geometry.

The following amendments are required for this manuscript.

Introduction should be expand. The authors listed references [7-15]. It is better to explain these works in concise manner in the introduction.

Eq. (1) is derived from Equation (64)? How is it possible?

Abbreviations should also be included in the nomenclature.

The conclusion section should be added in the manuscript.

Reviewer 2's comments:

The authors combine the radiation, convection, and conduction phenomenon using the single Monte Carlo algorithm. The algorithm would be beneficial for the complex geometries from computer graphics and its advances. The theoretical formulation has been presented and different thermal paths have been made on which heat is transferred. They probabilized Green’s function and propagators using a theoretical framework. Then they extended the model to make it operational by using the Feynman-Kac theory and stochastic processes. They presented Brownian trajectories and make them compatible with the algorithmic design required by the ray-tracing acceleration technique for highly refined geometries.

Following are my suggestion for this manuscript:

In different section of the article, the flow chart the sample computer code is written. I think there is a need of a complete flow chart that describe the algorthim start and end points by highlighting the coupling spotlights. The flow chart can also describe fluid and solid domains for heating pathways.I think there should be a portion that describe that why this type of coupling is required for a physical phenomenon and what kind of target audience would be beneficial.The introduction section is very brief also not including the work of other people. How the people have worked previously. How thery tried to couple this type of models. Research gap needs to be highlighted.The randomized sampling method should be explain more with kind of error analysis. So the path for next step should be determined correctly. The stepping method and size for the path needs more mathematical explanation.In figure 7b, the radiative path is making a triangle, is there a specific reason for the twist or sampling method is changing it? More explanation is required for changing path.Figure 7b can be made separately with an enlarge view.Equation 57 is an average of quantities. Why is it so? The exactness of this relation can be highlighted mathematically.==============================

We look forward to receiving your revised manuscript.

Kind regards,

Rab Nawaz

Academic Editor

PLOS ONE

Journal Requirements:

"This work received financial support from the French National Agency for Research

(ANR project HIGH-TUNE ANR-16-CE01-0010, ANR project MC2 ANR-21-CE46-0013

and ANR project MCG-RAD ANR-18-CE46-0012) and from Region Occitanie (Projet CLE EDSTAR). This work has also been sponsored by the French government research program ”Investissements d’Avenir” through the IDEX-ISITE initiative 16-IDEX-0001 (CAP 20-25), the IMobS3 Laboratory of Excellence (ANR-10-LABX-16-01) and the SOLSTICE laboratory of Excellence (ANR-10-LABX-22-01)"

"This work received financial support from the French National Agency for Research, https://anr.fr/, (ANR project HIGH-TUNE ANR-16-CE01-0010, ANR project MC2 ANR-21-CE46-0013 and ANR project MCG-RAD ANR-18-CE46-0012), from Region Occitanie (Projet CLE EDSTAR), from French government research program "Investissements d'Avenir" through the IDEX-ISITE initiative 16-IDEX-0001 (CAP 20-25), the IMobS3 Laboratory of Excellence (ANR-10-LABX-16-01) and the SOLSTICE laboratory of Excellence (ANR-10-LABX-22-01)."

4. "In your Data Availability statement, you have not specified where the minimal data set underlying the results described in your manuscript can be found. PLOS defines a study's minimal data set as the underlying data used to reach the conclusions drawn in the manuscript and any additional data required to replicate the reported study findings in their entirety. All PLOS journals require that the minimal data set be made fully available. For more information about our data policy, please see http://journals.plos.org/plosone/s/data-availability.

We will update your Data Availability statement to reflect the information you provide in your cover letter."

Additional Editor Comments (if provided):

Reviewers' comments:

Reviewer's Responses to Questions

**Comments to the Author**

1. Is the manuscript technically sound, and do the data support the conclusions?

Reviewer #1: Yes

Reviewer #2: Yes

2. Has the statistical analysis been performed appropriately and rigorously? 

Reviewer #1: Yes

Reviewer #2: Yes

3. Have the authors made all data underlying the findings in their manuscript fully available?

Reviewer #1: Yes

Reviewer #2: Yes

4. Is the manuscript presented in an intelligible fashion and written in standard English?

Reviewer #1: Yes

Reviewer #2: Yes

5. Review Comments to the Author

Reviewer #1: The authors developed single Monte-Carlo algorithm for radiation, conduction and convection process. The authors used the theoretical frame works of propagators and Green's functions which demonstrate that a coupled model involving different physical phenomena can be probabilized. They extended the Feynman-Kac theory and stochastic processes. They addressed that the coupled Brownian trajectories compatible with the algorithmic design required by ray-tracing acceleration techniques in highly refined geometry.

The following amendments are required for this manuscript.

Introduction should be expand. The authors listed references [7-15]. It is better to explain these works in concise manner in the introduction.

Eq. (1) is derived from Equation (64)? How is it possible?

Abbreviations should also be included in the nomenclature.

The conclusion section should be added in the manuscript.

Reviewer #2: 1) In different section of the article, the flow chart the sample computer code is written. I think there is a need of a complete flow chart that describe the algorthim start and end points by highlighting the coupling spotlights. The flow chart can also describe fluid and solid domains for heating pathways.

2) I think there should be a portion that describe that why this type of coupling is required for a physical phenomenon and what kind of target audience would be beneficial.

3) The introduction section is very brief also not including the work of other people. How the people have worked previously. How thery tried to couple this type of models. Research gap needs to be highlighted.

4) The randomized sampling method should be explain more with kind of error analysis. So the path for next step should be determined correctly. The stepping method and size for the path needs more mathematical explanation.

5) In figure 7b, the radiative path is making a triangle, is there a specific reason for the twist or sampling method is changing it? More explanation is required for changing path.

6) Figure 7b can be made separately with an enlarge view.

7) Equation 57 is an average of quantities. Why is it so? The exactness of this relation can be highlighted mathematically.

6. PLOS authors have the option to publish the peer review history of their article (what does this mean?). If published, this will include your full peer review and any attached files.

Reviewer #1: No

Reviewer #2: No

---

## [Editor Report · Decision Letter 1]

14 Mar 2023

Coupling radiative, conductive and convective heat-transfers in a single Monte Carlo algorithm: a general theoretical framework for linear situations.

PONE-D-22-29121R1

Dear Dr. Stéphane Blanco,

We’re pleased to inform you that your manuscript has been judged scientifically suitable for publication and will be formally accepted for publication once it meets all outstanding technical requirements.

Kind regards,

Rab Nawaz

Academic Editor

PLOS ONE

Additional Editor Comments (optional):

The authors have responded positively to the queries raised by the expert reviewers. Thus, the research article deserves to be publihsed in Plos One. 
---

## [Editor Report · Acceptance letter]

23 Mar 2023

PONE-D-22-29121R1 

Coupling radiative, conductive and convective heat-transfers in a single Monte Carlo algorithm: a general theoretical framework for linear situations. 

Dear Dr. Blanco:

I'm pleased to inform you that your manuscript has been deemed suitable for publication in PLOS ONE. Congratulations! Your manuscript is now with our production department. 

Kind regards, 

on behalf of

Dr. Rab Nawaz 

Academic Editor

PLOS ONE